# Broad geographic sampling reveals the shared basis and environmental correlates of seasonal adaptation in *Drosophila*

Heather E Machado[1,2†*], Alan O Bergland[1,3*], Ryan Taylor[1], Susanne Tilk[1], Emily Behrman[4], Kelly Dyer[5], Daniel K Fabian[6,7], Thomas Flatt[6,8], Josefa González[9], Talia L Karasov[10], Bernard Kim[1], Iryna Kozeretska[11,12], Brian P Lazzaro[13], Thomas JS Merritt[14], John E Pool[15], Katherine O'Brien[4], Subhash Rajpurohit[4], Paula R Roy[16], Stephen W Schaeffer[17], Svitlana Serga[11,12], Paul Schmidt[4‡*], Dmitri A Petrov[1‡*]

[1]Department of Biology, Stanford University, Stanford, United States; [2]Wellcome Sanger Institute, Hinxton, United Kingdom; [3]Department of Biology, University of Virginia, Charlottesville, United States; [4]Department of Biology, University of Pennsylvania, Philadelphia, United States; [5]Department of Genetics, University of Georgia, Athens, United States; [6]Institute of Population Genetics, Vetmeduni Vienna, Vienna, Austria; [7]Centre for Pathogen Evolution, Department of Zoology, University of Cambridge, Cambridge, United Kingdom; [8]Department of Biology, University of Fribourg, Fribourg, Switzerland; [9]Institute of Evolutionary Biology, CSIC- Universitat Pompeu Fabra, Barcelona, Spain; [10]Department of Biology, University of Utah, Salt Lake City, United States; [11]Taras Shevchenko National University of Kyiv, Kyiv, Ukraine; [12]National Antarctic Scientific Centre of Ukraine, Taras Shevchenko Blvd., Kyiv, Ukraine; [13]Department of Entomology, Cornell University, Ithaca, United States; [14]Department of Chemistry & Biochemistry, Laurentian University, Sudbury, Canada; [15]Laboratory of Genetics, University of Wisconsin-Madison, Madison, United States; [16]Department of Ecology and Evolutionary Biology, University of Kansas, Lawrence, United States; [17]Department of Biology, The Pennsylvania State University, University Park, United States

\*For correspondence:
heather.machado@sanger.ac.uk (HEM);
aob2x@virginia.edu (AOB);
schmidtp@upenn.edu (PS);
dpetrov@stanford.edu (DAP)

†These authors also contributed equally to this work
‡These authors also contributed equally to this work

Competing interests: The authors declare that no competing interests exist.

**Abstract** To advance our understanding of adaptation to temporally varying selection pressures, we identified signatures of seasonal adaptation occurring in parallel among *Drosophila melanogaster* populations. Specifically, we estimated allele frequencies genome-wide from flies sampled early and late in the growing season from 20 widely dispersed populations. We identified parallel seasonal allele frequency shifts across North America and Europe, demonstrating that seasonal adaptation is a general phenomenon of temperate fly populations. Seasonally fluctuating polymorphisms are enriched in large chromosomal inversions, and we find a broad concordance between seasonal and spatial allele frequency change. The direction of allele frequency change at seasonally variable polymorphisms can be predicted by weather conditions in the weeks prior to sampling, linking the environment and the genomic response to selection. Our results suggest that fluctuating selection is an important evolutionary force affecting patterns of genetic variation in *Drosophila*.

## Introduction

Fluctuations in the environment are an inescapable condition for all organisms. While many of these fluctuations are entirely unpredictable, some are predictable to a degree, including those that occur on diurnal and seasonal time scales. The predictability of cyclic fluctuations is reflected in the fact that many species exhibit plastic physiological and behavioral strategies that enable them to survive the unfavorable season and exploit the favorable one (*Denlinger, 2002*; *Kostál, 2006*); such plastic responses represent the classical form of seasonal adaptation (*Tauber et al., 1986*). However, seasonally varying selection can – in principle – maintain fitness-related genetic variation if some genotypes have high fitness in one season but not another (*Gillespie, 1973*; *Haldane and Jayakar, 1963*). Thus, in organisms that undergo multiple generations per year, a distinct form of seasonal adaptation occurs when the frequency of alternate genotypes changes in response to seasonal fluctuations in the environment.

Seasonal adaptation can be seen as a form of local adaptation – local adaptation in time to temporally varying selection pressures rather than in space. Such adaptation in time has often been considered to be uncommon and, when present, unlikely to result in long-term balancing selection (*Ewing, 1979*; *Hedrick, 1976*). Classic quantitative genetic theory suggests that an optimal, plastic genotype will eventually dominate a population that is exposed to periodically changing environments (*Bürger and Gimelfarb, 1999*; *Korol et al., 1996*; *Lande, 2008*; *Scheiner, 1993*). This is particularly so when certain environmental cues are reliable indicators of changes in selection pressure (*Levins, 1974*; *Via and Lande, 1985*). Predictions from traditional population genetic models suggest that periodically changing environments will lead to the rapid loss of seasonally favored ecotypes as slight changes in selection pressure from one year to another eventually push allele frequencies at causal alleles to fixation (*Hedrick, 1976*).

Recent theoretical models have called these classical predictions into question. For instance, a population genetic model by *Wittmann et al., 2017* has demonstrated that seasonally varying selection can indeed maintain fitness-related genetic variation at many loci throughout the genome provided that dominance shifts from season to season in such a way that, on average, the seasonally favored allele remains slightly dominant (*Curtsinger et al., 1994*). This model, along with others that highlight the importance of population cycles (*Bertram and Masel, 2019*), as well as overlapping generations and age structure (*Bertram and Masel, 2019*; *Ellner, 1996*; *Ellner and Hairston, 1994*; *Ellner and Sasaki, 1996*), suggest that seasonal adaptation and adaptive tracking (*Kain et al., 2015*) could be an important feature of organisms such as *Drosophila* that have multiple generations per year (*Behrman et al., 2015*, but see *Botero et al., 2015*). More generally, it is possible that adaptive tracking of environmental fluctuations on rapid time scales might be more common than generally acknowledged and has been hidden from us due to the difficulty of detecting such adaptive tracking reliably (*Lynch and Ho, 2020*) and from the lack of finely resolved temporal data (*Buffalo and Coop, 2020*).

Despite the lack of theoretical consensus on whether and how seasonal adaptation operates, there is substantial empirical evidence for seasonal adaptation in many organisms including *Drosophila*. Seasonal adaptation was first observed in *D. pseudoobscura* by Dobzhansky and colleagues (e.g., *Dobzhansky, 1943*) by tracking frequencies of inversion polymorphisms over seasons. Later studies confirmed and extended these early findings to other species including *D. melanogaster* (*Kapun et al., 2016*; *Stalker, 1980*; *Stalker, 1976*) and *D. subobscura* (*Rodríguez-Trelles et al., 2013*).

In *D. melanogaster*, multiple additional lines of evidence from phenotypic and genetic analyses demonstrate the presence of seasonal adaptation. When reared in a common laboratory environment, flies collected in winter or spring (close descendants of flies that successfully overwintered) show higher stress tolerance (*Behrman et al., 2015*), greater propensity to enter reproductive dormancy (*Schmidt and Conde, 2006*), increased innate immune function (*Behrman et al., 2018*), and modulated cuticular hydrocarbon profiles (*Rajpurohit et al., 2017*) as compared to flies collected in the fall (the descendants of those flies who prospered during the summer). Rapid adaptation over seasonal time scales in these and related phenotypes has also been observed in laboratory (*Schmidt and Conde, 2006*) and field-based mesocosm experiments (*Erickson et al., 2020*; *Grainger et al., 2021*; *Rajpurohit et al., 2018*; *Rajpurohit et al., 2017*; *Rudman et al., 2019*; Rudman and Greenblum et al., 2021). Genome-wide analysis indicated that a large number of

common polymorphisms change in frequency over seasonal time scales in one mid-latitude orchard (*Bergland et al., 2014*), and allele frequency change among seasons has been observed using candidate gene approaches (*Cogni et al., 2014*). In several cases, these adaptively oscillating polymorphisms have been linked to seasonally varying phenotypes (*Behrman et al., 2018*; *Erickson et al., 2020*; *Paaby et al., 2014*).

Despite ample evidence of seasonal adaptation in *D. melanogaster*, many aspects of this system remain unexplored. First, we do not know whether seasonal adaptation is a general feature of *D. melanogaster* populations across its range. Previous work (*Schmidt and Conde, 2006*; *Bergland et al., 2014*; *Cogni et al., 2014*; *Behrman et al., 2015*; *Behrman et al., 2018*) detected seasonal allele frequency fluctuations in a single locality over the span of 1–3 years. These results need to be confirmed at other locations. Second, as many ecological and environmental variables covary with season, it is unclear which specific factors drive the observed changes in phenotype and genotype in populations sampled over seasonal time. One straightforward and statistically powerful way to start answering these questions is to assess the extent of parallel shifts in allele frequencies over seasonal time in many populations across the *D. melanogaster* range. Should such parallel shifts be detected we can then infer that seasonal adaptation is widespread and at least partly driven by a common set of variants. We can also attempt to associate the magnitude and direction of allelic shifts with local environmental conditions (e.g., temperature) that may vary among populations sampled at equivalent points in seasonal time. Broad sampling will also allow us to determine the magnitude of the shared seasonal fluctuations and give us a glimpse into the genetic architecture of seasonal adaptation: for instance, whether or not the seasonally responsive variants are found genome-wide or clustered into linked blocks or even supergenes.

Here we carry out such a study by using allele frequency data from 20 paired seasonal samples from 15 localities in North America and Europe. First, we demonstrate that seasonal adaptation is a general phenomenon that occurs in multiple populations of *D. melanogaster* on two continents by providing evidence that at least some of the same polymorphisms cycle between seasons in populations sampled across vast geographic distances. Seasonal alleles tend to show clinal variation, with the alleles that increase in frequency through the summer generally being more frequent in lower latitude (more 'summer-like') locations. We also show that seasonal signal is enriched near inversion breakpoints or within inverted regions on chromosome 2L, 3L, and 3R, although these inverted regions do not account for all of the signal that we observe. Furthermore, we show that allele frequency change between seasons is most predictable when taking into account the local temperature prior to sample collections both in the spring and in the fall, which hints at the complex and nonlinear nature of adaptation with adaptive tracking occurring at sub-seasonal timescales. Taken together, our work demonstrates that seasonal adaptation is a general and predictable feature of *D. melanogaster* populations and has pervasive effects on patterns of allele frequencies genome-wide. More generally, we establish that metazoan populations can exhibit adaptive tracking of environmental conditions on extremely short time scales, most likely shorter than a single growing season encompassing ~10 generations.

## Results

### Fly samples and sequence data

We assembled 73 samples of *D. melanogaster* collected from 23 localities in North America and Europe (*Supplementary file 1*, *Figure 1—figure supplement 1*). For 15 sampling localities, flies were collected in the spring and fall over the course of 1–6 years (*Figure 1A*). Our sampling and sequencing efforts also complement other genome-wide datasets that investigate genetic variation within and among *D. melanogaster* populations throughout their range (*Campo et al., 2013*; *Fabian et al., 2012*; *Grenier et al., 2015*; *Kao et al., 2015*; *Kapun et al., 2020*; *Kim et al., 2014*; *Kolaczkowski et al., 2011*; *Lack et al., 2016*; *Mackay et al., 2012*; *Pool et al., 2012*; *Reinhardt et al., 2014*; *Svetec et al., 2016*; *Zhao and Begun, 2017*).

For our analysis, we divided our samples into two subsets (*Figure 1B*). The first subset ('Core20') is composed of 20 pairs of spring and fall samples from the same location. Samples in the Core20 set are drawn from 15 localities in North America and Europe and we use at most 2 years of sampling from any one locality. The second subset of samples ('Clinal') was used to examine patterns of

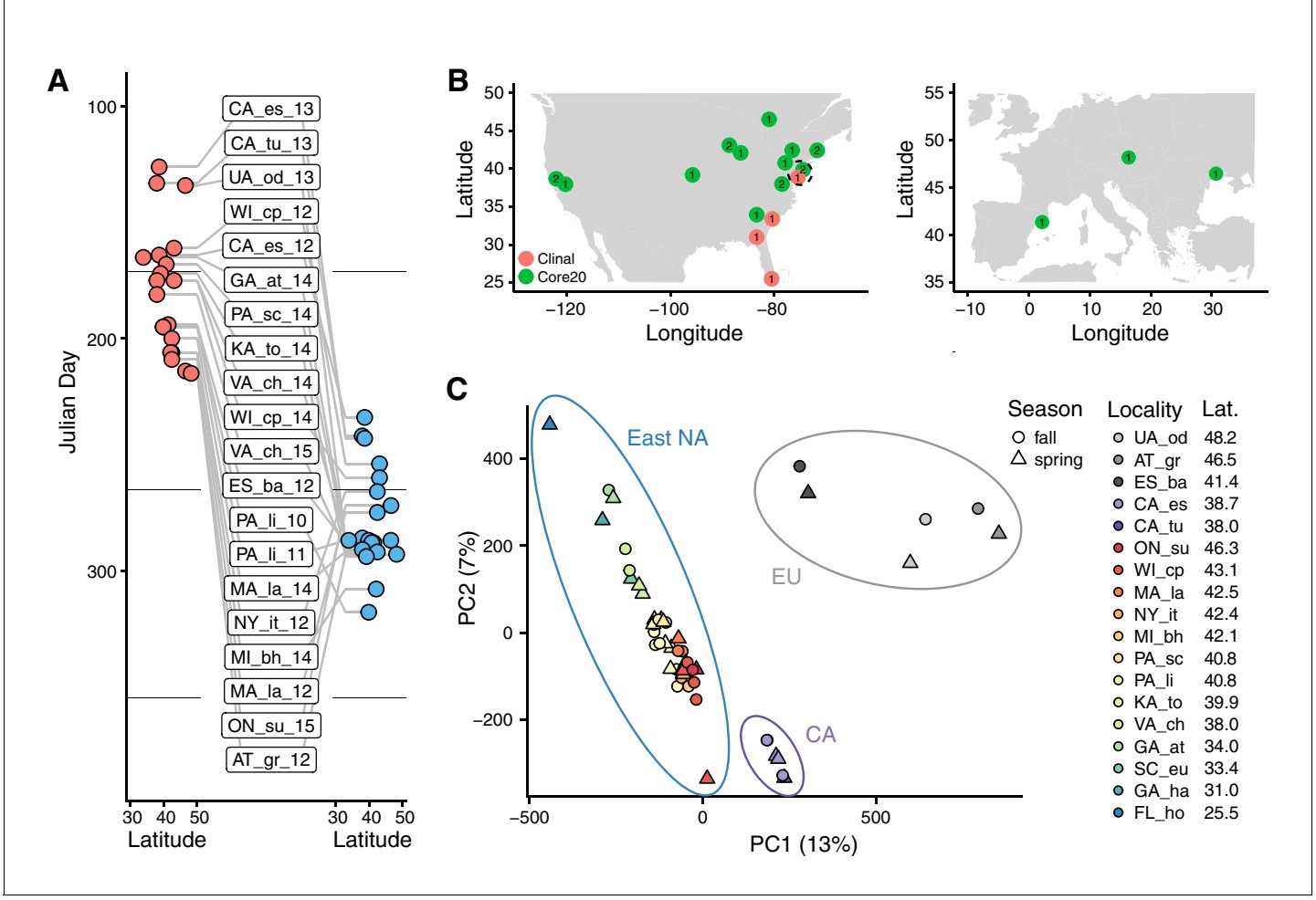

**Figure 1.** Sampling times, localities, and basic population structure of samples used in this study. (**A**) Distribution of collection times during the spring (red) and fall (blue) in relation to latitude. For samples where the collection month, but not day, was recorded, we specified the 15th of the month to calculate Julian Day. (**B**) Sampling localities for the primary seasonal analysis ('Core20': green) and the latitudinal cline analysis ('Clinal': red), distributed across North America and Europe. Numbers represent the number of years of spring/fall comparisons at that locality. The dotted circle shows the one locality (PA_li) with both Core20 and Clinal samples. (**C**) Principal component analysis of SNP allele frequencies. Circles are placed around samples from three geographic regions: Europe (EU), California (CA), and Eastern North America (East NA); points are colored by latitude and shapes represent seasons, as defined by collection time.

The online version of this article includes the following figure supplement(s) for figure 1:

**Figure supplement 1.** All sampling localities collected.

**Figure supplement 2.** Effect of filtering monomorphic variants on allele frequency distributions.

clinal variation along the East Coast of North America and consists of four populations sampled in the spring (see Materials and methods and *Supplementary file 1*).

To estimate allele frequencies genome-wide, we performed pooled DNA sequencing of multiple individual flies per population (*Schlötterer et al., 2014*; *Zhu et al., 2012*). For each sample, we resequenced pools of ~75 individuals (range 27–164) to an average of 94× coverage (range 22–220, *Supplementary file 1*). Analyses presented here use a total of 43 samples in the Core20 and Clinal sets described above. Data from the remaining 30 samples, including those that do not constitute paired spring/fall samples, are not included in our analyses here but nonetheless form a part of our larger community-based sampling and resequencing effort and are deposited with the rest of the data (NCBI SRA; BioProject Accession #PRJNA308584; accession numbers for each sample can be found in *Supplementary file 1*).

Pooled resequencing identified ~1.75M SNPs following quality control and filtering for read depth and average minor allele frequency (minor allele frequency > 0.01). We applied a second filtering and retained ~775,000 SNPs that have observed polymorphism in each population sampled. This biases us toward assessing fluctuations at higher frequency polymorphisms (*Figure 1—figure supplement 2*).

To gain insight into basic patterns of differentiation among the sampled populations, we performed a principal component (PC) analysis across all samples. Samples cluster by locality and geography (*Figure 1C*) with PC one separating European, North American West Coast, and East Coast populations, while PC two separates the eastern North America samples by latitude (linear regression $p = 3*10^{-15}$, $R^2 = 0.84$). No PC is associated with season, following correction for multiple testing, implying that the spatial differentiation dominates patterns of SNP variation across these samples.

## Seasonal adaptation is a general feature of *D. melanogaster* populations

Our first objective was to determine whether or not we find evidence of parallel seasonal cycling of allele frequencies across multiple populations. To identify a signal of parallel allele frequency change over time we assessed the per-SNP change in allele frequency between spring and fall among the Core20 set of populations using three complementary methods that rely on different statistical models. For all three methods, we contrast the observed signal to permutations in which we shuffled the seasonal labels within each spring–fall paired sample. With the exception of the seasonal labels within paired samples, these permutations have the advantage of retaining all of the features of the data including patterns of genetic covariation and all of the noise in the allele frequency estimation.

We first used a generalized linear model (GLM), regressing allele frequency at each SNP on season across all of the populations. We find that the observed genome-wide distribution of 'seasonal' p-values from this model is not uniform and there is a pronounced excess of SNPs with low p-values (*Figure 2A*). The observed data have a stronger seasonal signal than most permutations. For example, at a p-value of $~4*10^{-3}$, which corresponds to the top 1% (Supplemental Table 2), the observed data have a greater number of seasonal SNPs than 96.8% of the permutations (*Figure 2B*). This enrichment is robust to the choice of quantile (*Figure 2—figure supplement 1*). Seasonal SNPs identified by *Bergland et al., 2014* are slightly enriched among the top 1% of SNPs identified using the GLM, relative to matched genomic controls, after excluding the overlapping Pennsylvanian populations ($\log_2$ odds ratio $\pm$ SD = 0.59 $\pm$ 0.37, $p_{perm}$ = 0.0512). Our top set of seasonally variable SNPs shift in frequency by ~4–8% (*Figure 2C*). Taken at face value, this corresponds to effective selection coefficients of ~10–30% per season (~1–3% per generation assuming 10 generations per season).

Next, we performed a parallel analysis using a Bayesian model, Bayenv, which uses a population covariance matrix to account for structure among samples (*Günther and Coop, 2013*). We found an enrichment of seasonal SNPs at high test statistic values (quantile = 0.01: Bayes factor = 2.6), with a greater enrichment of seasonal SNPs compared with 94% of permutations (*Figure 2B*). The set of seasonal SNPs identified in the Bayenv analysis shows substantial overlap with the GLM set. Specifically, for the top 1% of SNPs in the GLM analysis, 20% are also in the top 1% of the Bayenv analysis, compared with a naive expectation of 1% (*Figure 2—figure supplement 2*).

Finally, we developed a method derived from combining ranked-normalized p-values from Fisher's exact tests that compares allele frequencies within each pair of seasonal samples (see Materials and methods, *Figure 2—figure supplement 3*). We again find a deviation from the expected test statistic distribution indicating an excess of seasonally varying polymorphisms (*Figure 2B*) with the observed enrichment of high scores (quantile = 0.01: $X^2$ = 32) to be greater than more than 98% of permutations. These three analyses, using entirely distinct statistical machinery and assumptions, together provide strong evidence of the genome-wide excess of seasonal SNPs indicating a robust signal of parallel seasonal adaptation across multiple populations.

We examined whether seasonally variable SNPs are enriched on any particular chromosome. To perform this test, we calculated the enrichment of seasonal SNPs per chromosome in the observed data relative to the permutations (*Figure 2D*). Chromosomes 2L and 3R have the greatest nominal enrichment of seasonal SNPs relative to permutations, with a mean increase of 30% ($p_{perm}$ = 0.06) and of 18% ($p_{perm}$ = 0.08), respectively. Enrichment is much lower for chromosomes 2R and 3L ($-2$

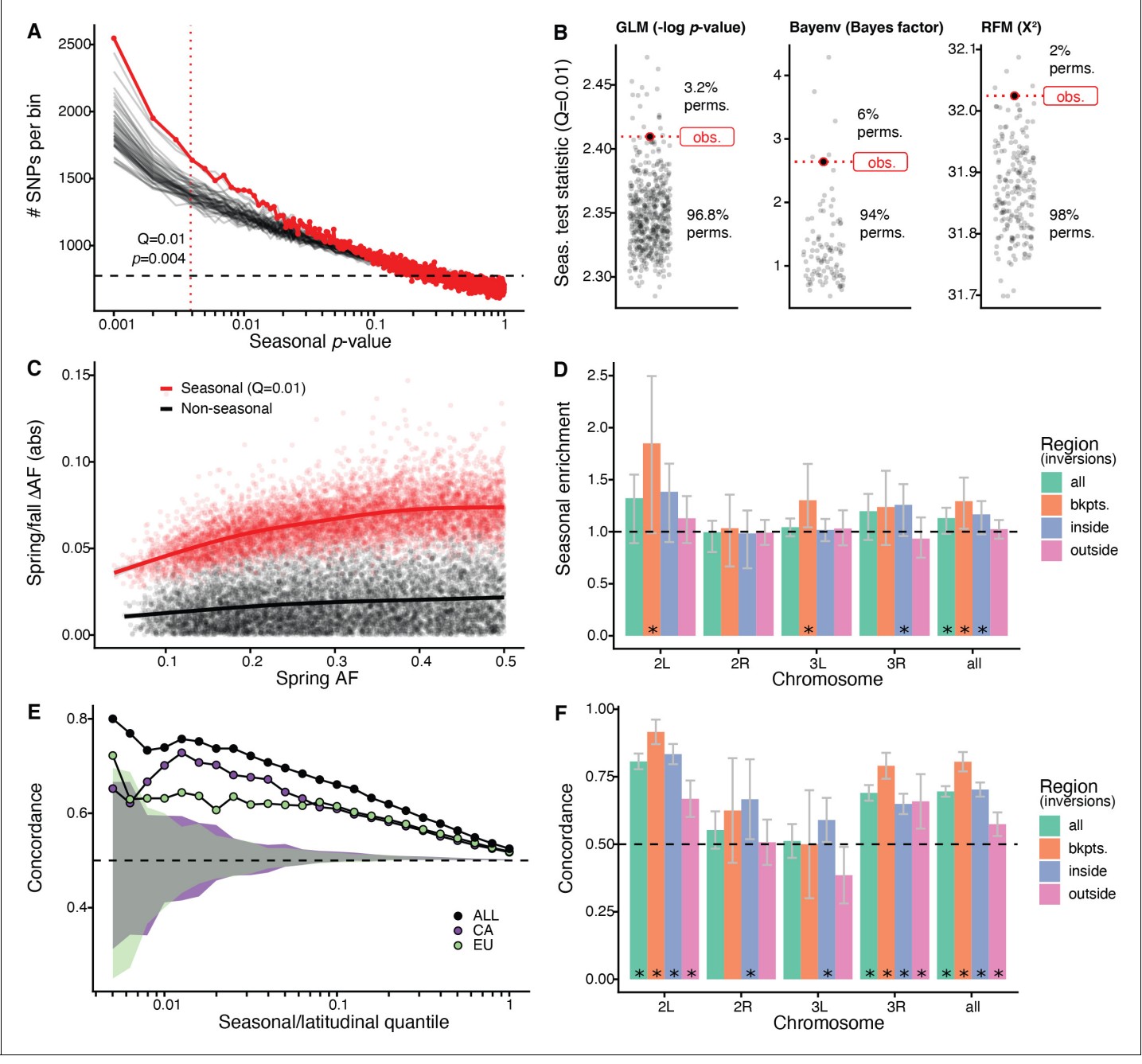

**Figure 2.** Signals of seasonal adaptation. (**A**) p-value distribution of GLM seasonal regression (red line), permutations (solid black lines, 50/500 plotted), and expected values (dashed black line). (**B**) Comparison of seasonal test statistics (quantile = 0.01) for observed (red) and permuted (black) datasets, for the three analyses: i: GLM (−log p-value), ii: Bayenv (Bayes factor; outlier of a Bayes factor of 10 excluded from plotting), iii: RFM ($X^2$). (**C**) Average spring/fall allele frequency change for each of the top 1% of seasonally varying SNPs (red) and non-seasonal, matched control SNPs (black), as a function of the folded spring allele frequency. Lines represent a moving average across SNPs with a given spring allele frequency. (**D**) Enrichment of seasonal SNPs (median of 500 permutations). Enrichment is calculated as the observed number of seasonal SNPs over the number of seasonal SNPs in each permutation at p<0.004. The genome is subset by chromosome and location relative to major cosmopolitan inversions (bkpts.: inversion breakpoints ± 1 Mb, inside: interior to the breakpoints excluding 1 Mb buffer, outside: exterior to the breakpoints excluding the 1 Mb buffers). Error bars are 95% confidence intervals and asterisks denote greater seasonal enrichment than more than 95% of permutations. (**E, F**) Concordance rates of seasonal and clinal polymorphisms by (**E**) geographic region and (**F**) genomic region. The y-axis represents the concordance rate of allele frequency change across time and space, assuming that the winter favored allele is the same as the allele favored in high latitudes. Clinal polymorphisms were identified along the East Coast of North America. Seasonal sites were identified using all populations not used in the clinal analysis (n = 18). (**E**) Seasonal sites were also identified using exclusively the California populations (purple: n = 3) or the Europe populations (green: n = 3). Shaded areas

*Figure 2 continued on next page*

*Figure 2 continued*

are the 95% confidence intervals of 100 matched control datasets for the Californian (purple) and European (green) analyses. (**F**) Concordance by genomic region. Error bars are 95% confidence intervals (binomial error) and asterisks denote concordance significantly greater than 0.5 (binomial test p<0.05).

The online version of this article includes the following figure supplement(s) for figure 2:

**Figure supplement 1.** Enrichment of seasonal SNPs by chromosome and significance threshold.
**Figure supplement 2.** Comparison of bayenv and GLM seasonal analyses.
**Figure supplement 3.** Artificial sample size increases for GLM and RFM methods.
**Figure supplement 4.** Comparison of the GLM and the Bayenv model for assessing clinal and seasonal concordance.

and 4%, $p_{perm}$ = 0.54 and $p_{perm}$ = 0.20, respectively). The variation of the seasonal signal across chromosomes is statistically significant (ANOVA, $p < 10^{-15}$).

As chromosomal inversions have been observed to fluctuate in frequency over seasonal time (*Dobzhansky, 1948*; *Dobzhansky, 1943*; *Kapun et al., 2016*), we extended the analysis to focus on large cosmopolitan inversions. We find that regions near inversion breakpoints (±0.5 Mb) on chromosomes 2L (85% enrichment, $p_{perm}$ = 0.03, inversion *In(2L)t*) and 3L (30% enrichment, $p_{perm}$ = 0.01, inversion *In(3L)Payne*) are strongly enriched for seasonal SNPs. Regions within the three inversions on chromosome 3R (*In(3R)Mo, In(3R)K, In(3R)Payne*) are also enriched for seasonal SNPs (26% enrichment, $p_{perm}$ = 0.05). We do not find evidence for enrichment of seasonal SNPs associated with *In(2R) NS* as previously reported by *Kapun et al., 2016*. The regions outside inversions (also excluding the 0.5 Mb flanking the inversions) represent the most centromeric/telomeric SNPs (36% of total SNPs). In these regions, the seasonal signal is weaker, with a 2.5% enrichment of seasonal SNPs ($p_{perm}$ = 0.31). These data confirm that inversions may play a role in the adaptive response to seasonally variable environments, similar to that of structural variants or large haplotype blocks in other systems (*Hager et al., 2021*; *Broad Institute Genome Sequencing Platform & Whole Genome Assembly Team et al., 2012*; *Samuk et al., 2017*; *Tuttle et al., 2016*).

## Latitudinal differentiation parallels signatures of seasonal adaptation

Parallel changes in phenotype and genotype across time and space have been observed in *D. melanogaster* (*Bergland et al., 2014*; *Cogni et al., 2015*; *Cogni et al., 2014*; *Kapun et al., 2016*; *Kapun and Flatt, 2019*; *Paaby et al., 2014*; *Rajpurohit et al., 2018*; *Rajpurohit et al., 2017*). To assess whether we observe similar patterns in our data, we tested whether SNPs that change in allele frequency along a latitudinal cline and across seasons do so in a parallel fashion. Parallel changes in allele frequency occur when the sign of the allele frequency change between spring and fall is the same as between high- and low-latitude populations. Using four populations along the East Coast latitudinal cline, we identified clinally varying polymorphisms using single-locus models (GLM, with allele frequency regressed against latitude) as well as a model that accounts for population structure (Bayenv; *Günther and Coop, 2013*). We re-identified seasonally varying SNPs using a GLM (as done previously) using 18 of the Core20 populations to avoid two samples drawn from a shared collection locale with the clinal set (PA_li; *Supplementary file 1*).

We find significant seasonal and latitudinal parallelism, with the magnitude of parallelism increasing with increasing GLM significance thresholds (*Figure 2E*). Among the polymorphisms that are in the top 1% of both seasonal and clinal GLMs (n = 115), 74% have parallel allele frequency change (greater than 100/100 matched controls; naive expectation: 50%). This pattern of parallelism is also observed when using a Bayenv model of clinality (*Figure 2—figure supplement 4*). Parallel changes in allele frequency between seasons and clines that we observe here is similar to previously published genome-wide (*Bergland et al., 2014*) and locus-specific results (*Cogni et al., 2014*; *Kapun et al., 2016*; *Stalker, 1980*; *Stalker, 1976*, p. 197).

To evaluate whether parallel allele frequency change across time and space are likely to be independent of shared demographic processes such as local seasonal migration, we compared East Coast clinal variation to seasonal variation identified in two groups of populations that are geographically and genetically distant from the East Coast (*Campo et al., 2013* and see *Figure 1B*) and that are unlikely to be connected via common seasonal migration. We assessed seasonal changes in allele frequency separately for Californian (n = 3) and European (n = 3) populations and tested for

parallelism with polymorphisms that vary along the East Coast of North America. We found a clear signal of parallelism between seasonal and latitudinal variation in both the California and the Europe comparisons: for the top 1% of seasonal and clinal polymorphism, 70% and 63% have parallel allele frequency change, for California and Europe, respectively (greater than 100/100 and 99/100 matched controls; *Figure 2E*).

As SNPs in or near a number of major inversions are enriched for seasonal variation (*Figure 2D*), we tested whether there is increased seasonal and latitudinal parallelism in these regions. For the top 5% of clinal and seasonal SNPs (n = 2132), parallel allele frequency change is highest near inversion breakpoints on chromosomes 2L and 3R (92%, $CI_{95}$: 87–96%, and 86%, $CI_{95}$: 78–93%, respectively) and inside the 2L inversion (83%, $CI_{95}$: 80–87%; *Figure 2F*). While parallelism is greatest overall around inversion breakpoints (concordance: 81%, $CI_{95}$: 77–84%) and inside inversions (concordance: 70%, $CI_{95}$: 67–73%), we also observe a statistically significant excess of latitudinal and seasonal parallelism outside of inversions (concordance: 57%, $CI_{95}$: 53–62%). The elevated signal of parallelism between spatial and seasonal allele frequency fluctuations across the whole genome, and in particular outside of inverted regions, suggests that inversions themselves, while important, are not the sole drivers of allele frequency change between seasons.

## Predictability of seasonal adaptation based on local environmental conditions

The signature of seasonal adaptation that we observe indicates that there are consistent changes in allele frequencies between seasons, broadly defined, among populations sampled across multiple years, and localities separated by thousands of miles (*Figure 1A,B*). Next, we asked if seasonal fluctuations in allele frequencies are predictable within any particular population or year and if the extent of predictability is related to aspects of local climate or other factors.

To address this question, we performed a leave-one-out analysis. In this analysis, we sequentially removed each of the 20 paired spring–fall populations within the Core20 set and re-identified seasonally variable SNPs among the remaining 19 populations as well as for the dropped, 20th population. To evaluate predictability, we calculated the proportion of SNPs that have the same direction of allele frequency change between spring and fall in both the discovery and test models (i.e., concordant allele frequency change). We estimated concordance across the range of joint significance thresholds of both models. For each population, we then calculated a 'predictability score'. This score reflects the change in concordance rate as a function of the joint significance threshold (see Materials and methods for details). Note that populations with a positive predictability score are those in which the concordance rate is greater than 50% over the bulk of the genome or chromosome.

We find that the majority of populations show a positive predictability score (*Figure 3A*). For these populations, the concordance rate reaches 60–70% among SNPs passing a stringent joint significance threshold (top 0.1% in both models). At more modest joint significance thresholds (e.g., the top 10% in both sets), the concordance rate is lower but still statistically different from 50% (maximum nominal p-value at the top 5% of sites is 0.008). At the same time, intriguingly, we found that four of the 20 populations had negative predictability scores (Benton Harbor, Michigan 2014; Lancaster, Massachusetts 2012; Topeka, Kansas 2014; and Esparto, California 2012). Strong allele frequency changes in these four populations occur, but the direction of allele frequency change from spring to fall at many SNPs is opposite that of the remaining populations.

We then asked why predictability scores vary among populations. To address this question we built models that related the genome-wide predictability score (*Figure 3A*) for each population with a number of different potential explanatory variables (*Figure 3B,C,D*). We first considered a number of nuisance variables such as collection method and substrate, as well as contamination rates by *D. simulans*. None of these variables are correlated with the genome-wide predictability score more than expected by chance (*Figure 3B*). Latitude and longitude are also not strong predictors of variation in genome-wide predictability score (*Figure 3B*).

We next focused on an environmental model based on aspects of the thermal environment. The motivation for these characterizations of temperature is based on the hypothesis that populations with negative genome-wide predictability scores were exposed to particularly warm springs and/or particularly cool falls prior to our sampling. This hypothesis is plausible considering that short-term changes in temperature have been linked, either directly or indirectly, to dramatic and partly

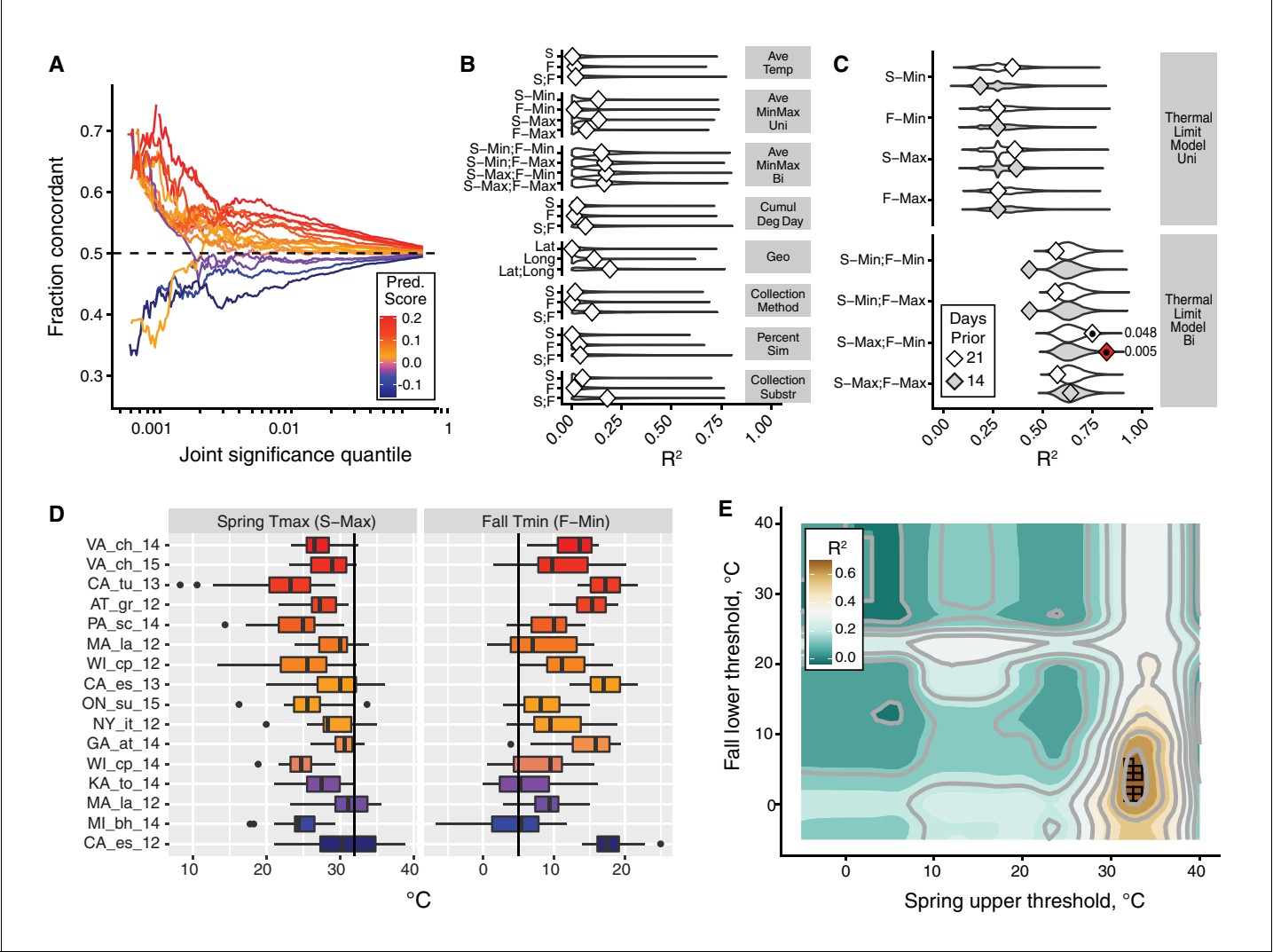

**Figure 3.** The thermal limit model. The thermal limit model argues that weather in the weeks prior to sampling can explain variation in the predictability of allele frequency change. (A) Each line represents the comparison between one population and the remaining 19. The x-axis represents the upper threshold of quantile-ranked p-values from the single-population test (Fisher's exact test) and the 19-population test (GLM), e.g., the top 1% in both tests. The y-axis is the fraction of SNPs where the sign of allele frequency change in the single population test matches the average sign change among the remaining 19. The color scheme represents the slope of this line and is used as a summary statistic for each population. (B, C). We regressed the summary score from (A) onto a number of characterizations of average temperature (B, first four rows), geography (B, fifth row), technical (B, sixth to 8th rows), and thermal extremes (C), considering weather 14. Diamonds represent the observed $R^2$ for (B) and observed maximum $R^2$ across all thermal limits for (C). Violin plots represent the expected distribution of $R^2$ based on permutations. The red diamond represents the model with nominal p-value<0.01. The empirical p-values for these models are listed next to the corresponding red diamond. The 14 day model that uses the counts of hot spring days and cold fall days has a false discovery rate of 17% based on multiple testing correction across all environmental models. (D) The distribution of spring maximum (S-Max) daily temperature and fall minimum (F-min) daily temperature in the 2 weeks prior to sampling. Discordant (blue) populations do not cluster in time or space. Populations shown here are those in which we have weather data (15 populations, in total). (E) The stand-out model uses the number of hot spring days and cold fall days. To determine the optimal threshold for what defines hot and cold, we systematically varied the upper and lower thermal limits from 0°C to 40°C and used the count of hot spring (x1) and cold fall (x2) days as independent, additive variables in a regression model; the genome-wide predictability score was used as the dependent variable (y). The best-fit model uses a spring max (S-Max) temp of ~32°C and a fall min (F-Min) temp of ~5°C and explains ~82% of the variation in the population predictability scores.

predictable changes in allele frequencies in the wild and in the laboratory in multiple Drosophilid species (*Barghi et al., 2019*; *Bergland et al., 2014*; *Mallard et al., 2018*; *Rodríguez-Trelles et al., 2013*; *Tobler et al., 2014*). We evaluated a number of features of temperature by calculating daily averages, average minimum and maximums, as well as the cumulative degree days (*Figure 3B*).

Finally, we calculated the number of days below and above thermal limits, with the thermal limits defined across a range of possible values (*Figure 3C*).

We find that the only set of features of the population that is significantly correlated with genome-wide predictability score is the number of hot spring days and number of cold fall days prior to sampling. The best fit thermal limit model explains 82.2% of the variation in genome-wide predictability scores (*Figure 3C and E*) and beats >99.5% of permutations (false discovery rate = 17%) (*Figure 3C*). Specifically, the thermal limits identified by this best-fit model are the number of days in the 2 weeks prior to spring sampling with maximum temperature above ~32°C and the number of days in the fall with minimum temperature below ~5°C. These values roughly correspond to upper and lower thermal limits for flies (*Hoffmann, 2010*; *Kellermann et al., 2012*; *Overgaard and MacMillan, 2017*). Taken together, these results suggest that exposure to temperature extremes, or to associated environmental variables, acts as a selective pressure in the wild, causing large fluctuations in allele frequency throughout the genome.

## Discussion

In this study, we have focused on the identification of genomic signatures of seasonal adaptation in *D. melanogaster*. Our approach relies on the detection of parallel changes in allele frequency from spring to fall across 20 populations. We use coarse definitions of spring and fall, with spring as the time of year close to when *D. melanogaster* first becomes locally abundant and fall as the time close to the end of the growing season but before populations become too scarce to sample (*Behrman et al., 2015*; *Ives, 1970*). We sampled populations across a wide geographic range, such that the length of the growing season as well as other environmental conditions varied substantially. Despite this environmental heterogeneity and the coarseness of the definitions of spring and fall, we detect clear signals of parallel seasonal allele frequency shifts across these populations (*Figure 2A, B*). Specifically, we demonstrate that seasonal adaptation is a general phenomenon of temperate *D. melanogaster* populations, and not restricted to a single orchard (Linvilla Orchard, Media, Pennsylvania: *Behrman et al., 2015*; *Bergland et al., 2014*; *Cogni et al., 2014*; *Schmidt and Conde, 2006*), biogeographic region (East Coast of North America: *Behrman et al., 2018*), or even a continent.

By detecting consistent seasonal changes in allele frequencies among multiple populations in North America and Europe, we can argue that seasonal adaptation across this whole range is at least partly driven by a consistent set of loci. Although parallel responses to selection in polygenic models of adaptation may be the exception rather than the norm due to redundancy (*Barghi et al., 2019*), we lack power to detect alleles that cycle in a small subset of populations or are private to a single population. We therefore focus only on the polymorphisms that segregate in all populations – and which consequently have a higher population frequency – and only look at the signal of parallel cycling. Thus, our conclusions are limited to the shared genetic basis of seasonal adaptation as revealed by the cycling of ~750,000 common SNPs which represent roughly half of all detected SNPs.

The detected seasonal SNPs can either be causal or, vastly more likely, linked to nearby causal genetic variants, be they SNPs, indels, structural variants, or transposable element insertions. We detected stronger seasonal signal near or inside inversions on chromosomes 2L, 3L, and 3R, suggesting that inversions are important drivers of seasonal adaptation (*Dobzhansky, 1948*; *Dobzhansky, 1943*; *Kapun et al., 2016*; *Kapun and Flatt, 2019*; *Rodríguez-Trelles et al., 2013*). Furthermore, the power to detect parallel seasonal evolution should be stronger for regions with reduced recombination, plausibly increasing the power of our approach to detect parallel adaptation linked to structural variants. Note that, by focusing on the subset of seasonal sites that also show clinal patterns of geographic differentiation, we were able to detect seasonal signal outside inversions as well. The cosmopolitan inversions span a substantial portion of the genome, and there is also likely to be substantial genetic diversity within these cytologically defined elements further complicating the inference. Thus, our study suggests that the complicated architecture of seasonal adaptation will require a much broader seasonal sampling of populations to fine-map the causal variants and that such an endeavor is not futile, given that it is possible to detect a set of loci that show consistent seasonal fluctuations with all the limitation of the approach notwithstanding. It is also clearly important in the future to obtain genomic data capable of identifying structural variation, including inversions, that plausibly underlie much of seasonal adaptation in *Drosophila*.

One key and poorly understood aspect of *Drosophila* seasonal adaptation is how the populations survive annual population collapse during the winter, manage to overwinter, and then manage to expand and recolonize the available resources during the summer (*Ives, 1970*; *Shpak et al., 2010*). Also poorly understood is the extent to which these patterns of collapse-overwintering-repopulation involve migration and refugia (*Coyne and Milstead, 1987*; *Ives, 1970*). However, it is clear that seasonal adaptation in *D. melanogaster* cannot involve wholesale population replacements by migration because the spatial structure of the *D. melanogaster* populations is robust year after year as evidenced by strong spatial population structure (*Figure 1C*) and persistent clinal patterns (*Machado et al., 2016*; *van Heerwaarden and Hoffmann, 2007*). While one can conceive of migration from local refugia playing a role, any such scenario still requires natural selection to operate in order to reset populations back to the 'spring' state every year. The full metapopulation dynamics and the ways that local *D. melanogaster* survive seasonal crashes and expansions remains to be understood.

Our ability to detect a shared genetic basis of seasonal adaptation across these broadly distributed populations implies that there are generic features in the environment that exert consistent selective pressures despite the environmental heterogeneity across sampling times and locations. It is possible that such generic selective pressures relate to temperature, humidity, population density, resource availability, or other biotic and abiotic factors (*Bogaerts-Márquez et al., 2021*). The seasonal and latitudinal parallelism in allele frequency change that we observe (*Figure 2E*) suggests that the environmental factors that drive these patterns must both shift between seasons locally and vary spatially from south to north.

While the exact nature of the environmental variables that drive seasonal selection remain to be determined, here we provide evidence that temperature extremes in the weeks prior to sampling are a strong proximate or even ultimate selective agent (*Figure 3*). Specifically, we show that the seasonal behavior of any one population could be statistically associated with temperature extremes in the 2 weeks prior to sampling both in the spring and in the fall. This model accounts for why several populations showed reversal of direction of movement of seasonal SNPs relative to the bulk of the populations; these populations had unusually warm springs and unusually cool falls prior to sampling. We thus hypothesize that, at the time of sampling, some of our spring samples had already experienced sufficiently strong selection in the generically summer way and that the fall samples had experienced selective pressures in the generically winter way. This would result in seasonal alleles in the spring already showing fall-like patterns and the ones in the fall showing spring-like patterns.

These findings also raise an intriguing possibility that natural selection may not push populations in the same consistent direction across the entire growing season but rather that natural selection and adaptation may be more heterogeneous through time (*Grant, 2002*; *Nosil et al., 2020*; *Siepielski et al., 2009*). In the future, more dense temporal sampling might help map the action of natural selection to more specific environmental factors and to quantify how fast the selective pressures change in strength and even direction across a single growing season. Our results thus illustrate that evolution and ecology do operate on similar timescales in this system (*Hendry, 2020*; *Schoener, 2011*; *Thompson, 1998*; *Yoshida et al., 2003*) and that evolutionary adaptation cannot be generally ignored a priori in demographic or ecological investigations even on extremely short timescales (*Rudman et al., 2018*).

While the number of detected seasonal loci is likely to be a vast overestimate of the number of causal seasonal variants, particularly if there is selection on large inversions, the strength of selection implied by the magnitude of fluctuations at seasonal sites is likely to be an underestimate. Partially linked loci should show a smaller magnitude of fluctuation than the true causal sites. Thus the 4–8% seasonal frequency shifts at the top 1% of the seasonal sites and the corresponding effective selection coefficients of ~10–30% per season (~1–3% per generation) imply even larger magnitudes of selection at the causal sites. Furthermore, given that we can only detect patterns of seasonal cycling that are at least partially consistent across all or most sampled populations, we are likely to underestimate the impact of linked seasonal selection due to selection that is occurring in a population-specific manner. While given our dataset it is impossible to know the true extent of polygenicity, our results suggest that seasonal adaptation acts on at least several but potentially a large number of dispersed loci affecting linked variation genome-wide. This seasonal selection force could rival in strength genetic draft due to selective sweeps and should be much stronger over seasonal timescales than random genetic drift in populations as large as *D. melanogaster* (*Karasov et al., 2010*).

Our observation of signals of seasonal adaptation at common SNPs demonstrates that polymorphism is maintained across the range of *D. melanogaster* despite strong fluctuating selection. The most straightforward explanation of these results is that these polymorphisms are subject to balancing selection (*Bertram and Masel, 2019*; *Charlesworth, 2015*; *Gulisija and Kim, 2015*, *Levene, 1953*; *Wittmann et al., 2017*). Pervasive balancing selection is consistent with the recent realization that simple models of mutation-selection balance are inconsistent with the extent of quantitative genetic variation in a variety of fitness-related traits (*Charlesworth, 2015*). A full understanding of the genomic consequences of balancing selection caused by strong fluctuating selection will require the identification of individual causal loci. Doing so will require substantial additional sampling across seasons as well as from closely and broadly distributed locales throughout the whole geographic range of *D. melanogaster* in combination with the genetic mapping of seasonally fluctuating phenotypes (*Behrman et al., 2018*; *Behrman et al., 2015*; *Erickson et al., 2020*; *Rajpurohit et al., 2018*; *Rajpurohit et al., 2017*; *Schmidt and Conde, 2006*). Finally, such efforts can be strongly augmented by manipulative field experiments and functional work to investigate the molecular effects of putatively causal genetic variants. All of this work will need to be done in the context of other processes that underlie persistence of populations in the face of environmental heterogeneity such as phenotypic plasticity and bet hedging (*Ayrinhac et al., 2004*; *Bergland et al., 2008*; *Kain et al., 2015*; *MacMillan et al., 2015*; *Overgaard et al., 2011*; *Via and Lande, 1985*). Overall, it is becoming clear that all of the key ecological and evolutionary processes, including demographic changes, mutation, migration, phenotypic plasticity, and rapid evolution by natural selection, can and do occur on very short and overlapping timescales and thus must be investigated as part of one complex evolving and interacting system.

## Materials and methods

### Population sampling and sequence data

We assembled 73 samples of *D. melanogaster*, 61 representing newly collected and sequenced samples, and 12 representing previously published samples (*Bergland et al., 2014*; *Kapun et al., 2016*). Locations, collection dates, number of individuals sampled, and depth of sequencing for all samples are listed in *Supplementary file 1*. For each sample, members of the *Drosophila* Real-Time Evolution Consortium (DrosRTEC) collected an average of 75 male flies using direct aspiration from substrate, netting, or trapping at orchards and residential areas. Flies were confirmed to be *D. melanogaster* by examination of the male genital arch. We extracted DNA by first pooling all individuals from a sample, grinding the tissue together in an extraction buffer, and using a lithium chloride – potassium acetate extraction protocol (see *Bergland et al., 2014* for details on buffers and solutions). We prepared sequencing libraries using a modified Illumina protocol (*Bergland et al., 2014*) and Illumina TrueSeq adapters. Paired-end 125 bp libraries were sequenced to an average of 94× coverage either at the Stanford Sequencing Service Center on an Illumina HiSeq 2000 or at the Stanford Functional Genomics facility on an Illumina HiSeq 4000.

The following sequence data processing was performed on both the new and the previously published data. We trimmed low-quality 3' and 5' read ends (sequence quality < 20) using the program cutadapt v1.8.1 (*Martin, 2011*). We mapped the raw reads to the *D. melanogaster* genome v5.5 (and for *D. simulans* genome v2.01, flybase.org) using bwa v0.7.12 mem algorithms, with default parameters (*Li and Durbin, 2009*), and used the program SAMtools v1.2 for bam file manipulation (functions index, sort, and mpileup) (*1000 Genome Project Data Processing Subgroup et al., 2009*). We used the program picard v2.0.1 to remove PCR duplicates (http://picard.sourceforge.net) and the program GATK v3.2–2 for indel realignment (*McKenna et al., 2010*). We called SNPs and indels using the program VarScan v2.3.8 using a p-value of 0.05, minimum variant frequency of 0.005, minimum average quality of 20, and minimum coverage of 10 (*Koboldt et al., 2012*). We filtered out SNPs within 10 bp of an indel (they are more likely to be spurious), variants in repetitive regions (identified by RepeatMasker and downloaded from the UCSC Genome browser), SNPs with a median frequency of less than 1% across populations, regions with low recombination rates (~0 cM/Mb; *Comeron et al., 2012*), and nucleotides with more than two alleles. Because we sequenced only male individuals, the X chromosome had lower coverage and was not used in our analysis. After filtering, we had a total of 1,763,522 autosomal SNPs. This set was further filtered to include only

SNPs found polymorphic in all samples ('common polymorphisms'), resulting in 774,651 SNPs that represent the core set used in our analyses.

Due to the phenotypic similarity of the species *D. melanogaster* and *D. simulans*, we tested for *D. simulans* contamination by competitively mapping reads to both genomes. We omitted any of our pooled samples with greater than 5% of reads mapping preferentially to *D. simulans* (Charlottesville, VA 2012, fall; Media, PA 2013 fall). For the remaining samples, reads that mapped better to the *D. simulans* were removed from the analysis. For the populations used in this analysis, the average proportion of reads mapping preferentially to *D. simulans* was less than 1% (see *Supplementary file 1*), and there was no significant difference in the level of *D. simulans* contamination between spring and fall samples (t-test p=0.90).

In silico simulation analysis shows that competitive mapping accurately estimates the fraction of *D. simulans* contamination. To conduct these simulations, we used a single *D. simulans* genome from an inbred line derived from an individual caught California (*Signor, 2017*) and a single *D. melanogaster* genome from a DGRP line (*Mackay et al., 2012*). We used wgsim to generate simulated short reads from each genome and mixed these short reads together in various proportions. We then mapped the short reads back to the composite genome for competitive mapping, as described above. We calculated contamination rate as the number of total reads mapping to the *D. simulans* reference genome divided by the number of reads mapping to both *D. simulans* and *D. melanogaster*. These simulations demonstrate that the estimation of the cross species mapping is precise (Pearson's r = 0.9999, p=$1.6 \times 10^{-24}$), but underestimates the true contamination rate by ~2%.

To assess seasonal variation, we analyzed population genomic sequence data from 20 spring and 20 fall samples ('Core20'). These samples represent a subset of the sequenced samples. We used samples that had both a spring and a fall collection taken from the same locality in the same year. We also used a maximum of 2 years of samples for a given locality to prevent the analysis from being dominated by characteristics of a single population. When there were more than 2 years of samples for a given population, we chose to use the 2 years with the earliest spring collection time. This decision was made on the assumption that the earlier spring collection would better represent allele frequencies following overwintering selection. This left 20 paired spring/fall samples, taken from 12 North American localities spread across 6 years and 3 European localities across 2 years (*Supplementary file 1*). The localities and years of sampling are as follows: Esparto, California 2012 and 2013; Tuolumne, California 2013; Lancaster, Massachusetts 2012 and 2014; Media, Pennsylvania 2010 and 2011; Ithaca, New York 2012; Cross Plains, Wisconsin 2012 and 2014; Athens, Georgia 2014; Charlottesville, Virginia 2014 and 2015, State College, Pennsylvania 2014; Topeka, Kansas 2014; Sudbury, Ontario, Canada 2015; Benton Harbor, Michigan 2014, Barcelona, Spain 2012; Gross-Enzersdorf, Vienna 2012; Odesa, Ukraine 2013. For comparison of seasonal with latitudinal variation, we used sequence data from four spring samples along the east coast of the United States (Homestead, Florida 2010; Hahia, Georgia 2008; Eutawville, South Carolina 2010, Linvilla, Pennsylvania 2009). We performed a principal component analysis of allele frequency per SNP per sample using the prcomp function in R.

## Identification of seasonal sites

We identified seasonal sites using three separate methods: a general linear regression model (GLM), a Bayesian model (Bayenv; *Coop et al., 2010*; *Günther and Coop, 2013*), and a model based on the rank Fisher's method (RFM). Statistical analyses were performed in R (*R Development Core Team, 2014*).

To perform the GLM we used the glm function with binomial error, weighted by the 'effective coverage' per SNP per population ($N_c$) – a measure of the number of chromosomes sampled, adjusted by the read depth:

$$N_c = (1/N + 1/R) - 1$$

where N is the number of chromosomes in the pool and R is the read depth at that site (*Kolaczkowski et al., 2011*; *Feder et al., 2012*; *Bergland et al., 2014*; *Machado et al., 2016*). This adjusts for the additional error introduced by sampling of the pool at the time of sequencing. The seasonal GLM is a regression of allele frequency by season (e.g., spring versus fall): $y_i$ = season + population+$e_i$ where $y_i$ is the allele frequency at the ith SNP, population is the unordered categorical

variable denoting each of the 20 populations, and ei is the binomial error at the ith SNP. Although the GLM is a powerful test, the corrected binomial sampling variance ($N_c$) is likely an anti-conservative estimate of the true sampling variance associated with pooled sequencing (*Machado et al., 2016*). Therefore, we use the results of this test as a seasonal outlier test, rather than an absolute measure of the deviation from genome-wide null expectation of no allele frequency change between seasons. Results are compared to 500 permutations of the spring and fall labels within each population pair.

We tested for signals of seasonality using Bayenv. First, we calculated the genetic-covariance matrix between the Core20 set. Next, we calculated Bayes Factor scores for each SNP using the observed spring and fall labels. We used the read counts of reference and alternate based on the corrected binomial sampling variance ($N_c$). We conducted 100 permutations of the spring and fall labels within each population pair.

Finally, we developed a test to identify signals of seasonality which we call the 'RFM' that is robust to the inflated sample size estimates that can arise when using pooled allele frequencies (*Machado et al., 2016*). For this method, we first perform a Fisher's exact test for spring–fall allele frequency differentiation within each population pair. We then rank-normalize the resulting p-values for any given population by grouping SNPs into categories based on the number of total number of reads and the number of alternate reads, summed between spring and fall. The class-based rank of each SNP becomes its new p-value, providing uniform sets of p-values across the genome. As the power (minimum p-value) is relative to the number of SNPs being ranked per bin, and as the bin sizes are equivalent across populations, each population has equivalent power. Additionally, we find that this method is robust to inaccurate estimates of allele frequency precision (*Figure 2—figure supplement 3*). We then combine ranked p-values using Fisher's method for each SNP, taking two times the negative sum of logged p-values across the 20 spring/fall comparisons (each tail tested separately). The null distribution for this statistic ($X^2$) is a chi-squared distribution with degrees of freedom equal to 40 (two times the number of comparisons). We used the distribution of these per-SNP $X^2$ test statistics to test for a seasonal signal above noise and compared to 200 spring/fall permutations.

## Inversions

We analyzed seasonal variation associated with six major cosmopolitan inversions found in *D. melanogaster*: *In(2L)t* (2L:2225744–13154180), *In(2R)NS* (2R:11278659–16163839), *In(3L)P* (3L:3173046–16301941), *In(3R)K* (3R:7576289–21966092), *In(3R)Mo* (3R:17232639–24857019), and *In(3R)P* (3R:12257931–20569732). As the three inversions on 3R are overlapping, we combined them. We classified the genome into three categories: inversion breakpoints (500 kb ± each inversion breakpoint; 104 k SNPs), inside inversions (excluding breakpoint regions; 390 k SNPs), and outside inversions (excluding breakpoint regions; 281 k SNPs).

## Matched controls

With the assumption that the majority of the genome is not seasonal, we can use matched genomic controls as a null distribution to test for enrichment of different features of seasonal polymorphisms. The use of matched control SNP sets was employed for the tests of seasonal and latitudinal parallelism. We matched each SNP identified as significantly seasonal (at a range p-values) to another SNP, matched for chromosome, effective coverage, median spring allele frequency (across populations), inversion status either within or outside of the major inversions *In(2L)t*, *In(2R)NS*, *In(3L)P*, *In(3R)K*, *In(3R)Mo*, and *In(3R)P*, and recombination rate. We used the same *D. melanogaster* inversion breakpoints used in *Corbett-Detig and Hartl, 2012* and the recombination rates from *Comeron et al., 2012*. We randomly sampled 100 of the possible matches per SNP (excluding the focal SNP) to produce 100 matched control sets. Any SNPs with fewer than 10 possible matches were discarded from the matched control analyses. We defined 95% confidence intervals from the matched controls as the 3rd and 98th ranked values for the quantity being tested (e.g., percent concordance or proportion of genic classes).

## Latitudinal cline concordance

To identify SNPs that changed consistently in allele frequency with latitude (clinal), we first identified SNPs that varied in allele frequency along a 14.4° latitudinal transect up the east coast of the United States. We used one spring sample from each of the following populations (identified as 'Region_City_Year'): PA_li_2011 (39.9°N), SC_eu_2010 (33.4°N), GA_ha_2008 (30.1°N), and FL_ho_2010 (25.5°N).

First, we regressed allele frequency with population latitude in a general linear model (glm: R v3.1.0), using a binomial error model and weights proportional to the effective coverage ($N_c$): $y_i = latitude + e_i$ where $y_i$ is the allele frequency at the ith SNP, and $e_i$ is the binomial error given the ith at the SNP. To confirm the robustness of the clinal GLM results, clinality also was assessed using a second, more complex model that accounts for existing covariance in the dataset (Bayenv: *Coop et al., 2010*; *Günther and Coop, 2013*).

We then tested the concordance in allele frequency change by season with the allele frequency change by latitude. We performed three separate seasonal regressions (see above) for comparison with the latitudinal regression: spring verses fall for the 18 non-Pennsylvania paired samples, spring versus fall for the three California paired samples, and spring versus fall for the three Europe paired samples. With the removal of the Pennsylvania samples, none of these three seasonal regressions contained samples from any of the four populations used for the latitudinal regression. Taking sets of increasingly clinal and increasingly seasonal SNPs, we assessed the proportion of sites that both increase in frequency from spring to fall and increase in frequency from north to south or that decrease in frequency from spring to fall and decrease in frequency from north to south. We compared this concordance with the concordance of 100 sets of matched controls.

## Predictability analysis

To test the general predictability of seasonal change in our dataset, we performed a leave-one-out analysis. In this analysis, we performed seasonal regressions for subsets of the data, dropping one paired sample and comparing it to a seasonal test of the remaining 19. For the single population model, we used a Fisher's exact test. For the multi-population model, we used a GLM. For either model, we used the effective number of reads ($N_c$, defined above). We then measured the percent concordance of spring/fall allele frequency change, defined as the proportion of SNPs that agree in the direction of allele frequency change between spring and fall. This was performed 20 times, once for each paired sample.

To estimate the predictability scores, we calculated the rate of change of the concordance score as a function of quantile threshold. To estimate this rate, we used the GLM with binomial error, $concordance_i = quantile_i + e_i$, where 'concordance' is the fraction of SNPs falling below the ith quantile threshold of both the target (19 population) and test (1 population) models that changed in frequency in a concordant fashion and e is the binomial error with weights corresponding to the total number of SNPs falling below that threshold. Thus, the genome-wide (or chromosome specific) predictability score is heavily influenced by concordance rates of higher quantiles because that is where the bulk of SNPs reside.

We tested whether the variation in genome-wide predictability scores among populations is correlated with different features of the samples. We evaluated different nuisance parameters, such as collection method, substrate, and *D. simulans* contamination rate, as well as a number of features related to temperature. Daily minimum and maximum temperatures were obtained from the Global Historical Climatology Network-Daily (GHCND) database (*Menne et al., 2012*). We matched each locality to a weather station based on geographic distance. For the Core20 set of populations, four of the collections did not have precise collection dates for one or both seasonal collections (Athens, Georgia 2014; Media, Pennsylvania 2010, 2011; Barcelona, Spain 2012) or were not associated with any climate data from the GHCND database (Odessa, Ukraine 2013). These five populations were omitted from the analysis, leaving 15 populations for the remainder of the predictability analysis (*Figure 3D*). Using daily temperature data, we calculated the cumulative growing degree days following a model developed for *D. suzukii* by *Wiman et al., 2014*.

We regressed the predictability score onto each potential explanatory variable, recorded the proportion of variation explained ($R^2$), and contrasted the observed $R^2$ to null distributions generated via permutation (n = 25,000). For most variables, we performed a univariate model (e.g., latitude or

longitude). In some cases, we used a bivariate model (e.g., latitude + longitude). To test the thermal limit model, we regressed the observed predictability score on the number of days above and below the thermal limits. We varied thermal limits from $-5$ to $40°C$ with a step size of $0.1°C$.

## Acknowledgements

We thank the National Evolutionary Synthesis Center (NESCent) for sponsoring the 2012 Catalysis meeting that initiated the *Drosophila* Real Time Evolution Consortium. The meeting was attended by Alan Bergland, Alisa Sedghifar, Brian Helmuth, Brian Lazzaro, Chau-Ti Ting, David Kidd, Dmitri Petrov, Fabian Staubach, Hannah Burrack, Jim Fry, John Lessard, John Coulbourne, John Pool, Josefa Gonzalez, Julien Ayroles, Kelly Dyer, Kim Hughes, Maaria Kankare, Nadia Singh, Paul Schmidt, Regan Early, Stephen Porder, Subhash Rajpurohit, Sui Fai Lee, and Thomas Flatt, and we kindly thank all participants for their participation. We also thank Jamie Blundell and all members of the Schmidt and Petrov labs who provided exceptionally valuable feedback.

## Additional information

### Funding

| Funder | Grant reference number | Author |
|---|---|---|
| NIH Office of the Director | R01GM100366 | Paul Schmidt<br>Dmitri A Petrov |
| NIH Office of the Director | R35GM118165 | Dmitri A Petrov |
| NIH Office of the Director | R01GM137430 | Paul Schmidt |
| NIH Office of the Director | F32GM097837<br>R35GM119686 | Alan O Bergland |
| European Commission | H2020-ERC-2014-CoG-647900 | Josefa González |
| Natural Sciences and Engineering Research Council of Canada | RGPIN-2018-05551 | Thomas Merritt |
| Canada Research Chairs | 950-230113 | Thomas Merritt |

The funders had no role in study design, data collection and interpretation, or the decision to submit the work for publication.

### Author contributions

Heather E Machado, Conceptualization, Data curation, Formal analysis, Investigation, Visualization, Methodology, Writing - original draft, Writing - review and editing; Alan O Bergland, Conceptualization, Resources, Formal analysis, Funding acquisition, Investigation, Visualization, Methodology, Writing - original draft, Project administration, Writing - review and editing; Ryan Taylor, Data curation, Formal analysis; Susanne Tilk, Investigation; Emily Behrman, Kelly Dyer, Daniel K Fabian, Thomas Flatt, Josefa González, Talia L Karasov, Iryna Kozeretska, Brian P Lazzaro, Thomas JS Merritt, John E Pool, Katherine O'Brien, Subhash Rajpurohit, Paula R Roy, Stephen W Schaeffer, Svitlana Serga, Resources; Bernard Kim, Formal analysis; Paul Schmidt, Conceptualization, Formal analysis, Investigation, Resources, Funding acquisition, Writing - original draft, Project administration, Writing - review and editing; Dmitri A Petrov, Conceptualization, Supervision, Formal analysis, Investigation, Funding acquisition, Writing - original draft, Project administration, Writing - review and editing

### Author ORCIDs

Heather E Machado (iD) https://orcid.org/0000-0002-1523-3937
Alan O Bergland (iD) https://orcid.org/0000-0001-7145-7575
Ryan Taylor (iD) https://orcid.org/0000-0002-9003-6378
Susanne Tilk (iD) http://orcid.org/0000-0002-9156-9360
Emily Behrman (iD) http://orcid.org/0000-0002-2472-9635

Thomas Flatt  http://orcid.org/0000-0002-5990-1503
Bernard Kim  https://orcid.org/0000-0002-5025-1292
Thomas JS Merritt  https://orcid.org/0000-0002-4795-7534
Katherine O'Brien  http://orcid.org/0000-0003-4660-0338
Dmitri A Petrov  https://orcid.org/0000-0002-3664-9130

## Decision letter and Author response

Decision letter https://doi.org/10.7554/eLife.67577.sa1
Author response https://doi.org/10.7554/eLife.67577.sa2

# Additional files

## Supplementary files
- Supplementary file 1. Sample metadata.

- Transparent reporting form

## Data availability

All raw sequence data have been deposited to the NCBI short read archive (SRA; BioProject Accession #PRJNA308584; accession numbers for each sample can be found in Supplementary file 1). Code to conduct these analyses, primary results files, and code to reproduce the figures are available at https://github.com/machadoheather/dmel_seasonal_RTEC (copy archived at https://archive.softwareheritage.org/swh:1:rev:c4e0bc07a642c35470cd2c21b4f38f7ed0daa28d). VCF files with the raw allele frequencies per population and a R-data file of allele frequencies and effective sample sizes (Nc; compatible with scripts) are available on DataDryad (https://doi.org/10.5061/dryad.4r7b826).

The following datasets were generated:

| Author(s) | Year | Dataset title | Dataset URL | Database and Identifier |
|---|---|---|---|---|
| Bergland AO | 2016 | Real Time Evolution Consortium: Tracking the tempo and mode of evolution over ecologically relevant temporal and spatial scales | https://www.ncbi.nlm.nih.gov/bioproject/PRJNA308584/ | NCBI BioProject, PRJNA308584 |
| Machado HE | 2021 | Data from: Broad geographic sampling reveals predictable, pervasive, and strong seasonal adaptation in Drosophila | https://doi.org/10.5061/dryad.4r7b826 | Dryad Digital Repository, 10.5061/dryad.4r7b826 |

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
