## [Decision Letter]

**Acceptance summary:**

This paper shows convincing evidence of correlated seasonal changes in allele frequency over large geographical scales.

**Decision letter after peer review:**

[Editors’ note: the authors submitted for reconsideration following the decision after peer review. What follows is the decision letter after the first round of review.]

Thank you for submitting your work entitled "Broad geographic sampling reveals predictable, pervasive, and strong seasonal adaptation in *Drosophila*" for consideration by *eLife*. Your article has been reviewed by 3 peer reviewers, including Magnus Nordborg as the Reviewing Editor and Reviewer #3, and the evaluation has been overseen by a Senior Editor. The following individual involved in review of your submission has agreed to reveal their identity: Jeffrey Ross-Ibarra (Reviewer #2).

Our decision has been reached after consultation between the reviewers. Based on these discussions and the individual reviews below, we regret to inform you that your work will not be considered further for publication in *eLife* at this stage.

This decision is not based on the quality of the study, or of the importance of the work, but rather on problems with the analysis, which we think will require months to rectify". Therefore, our decision should be seen as "reject, but encourage to resubmit".

As is evident from the individual reviews below, there is no doubt that this is an important and interesting study. However, while we are generally inclined to believe most of your results, the quality of the analysis leaves much to be desired. It needs to be simplified, and the many ad hoc assumptions justified or dropped.

There are also serious concerns about sampling (reviewer #1) and about whether the proposed model would actually work.

We hope these comments are helpful.

*Reviewer #1:*

In this manuscript the authors present an analysis of seasonal allele frequency change in multiple population samples of *D. melanogaster* that have been sampled in spring and fall from a number of collection sites within North America and Europe. Briefly, the authors search for seasonality of allele frequencies using simple linear models and then look for coordinated changes among populations. The authors report decent overlap among "seasonal" snps, overlap with sites found to be clinal, as well as a number of ancillary results.

In general I think the dataset is quite interesting and the study to be timely. While that is so there are a number of technical issues throughout the paper and it suffers in numerous places from what I think are inappropriate statistical assumptions. While this is so I'm sympathetic to what the authors are attempting to do-it is hard to do correctly.

1) I have a number of concerns about the sampling design of this study:

a. First and foremost there is no consistency to the sampling date between years or localities. For instance the Fall samples from Linvilla, PA were collected either in October or November. Indeed a quick search for average temps reveals that at their collection site the average daily temperature differs by ~ 10 degrees F between those two dates. Thus I'm concerned that "season" is not a properly replicated factor across the design and instead there could be significant batch effects.

b. I'm concerned about the possibility of contamination from accidental collection of D. simulans that should vary systematically between the seasons (we know that the abundance of sim vs mel in local collections changes dramatically throughout the season. While the authors are rightly trying to account for this by competitive mapping more assurances should be given that this is working. I would like to see a simple simulation demonstrating the power of the competitive mapping proceedure-a straightforward way to do this would be to make synthetic pools of reads from *melanogaster* and simulans, while varying the percentage of simulans in the pool, and then take those simulated pools through the authors' mapping procedure. In addition, the authors should provide the percentage of reads that mapped to the sim assembly for each sample in Suppl Table 1. Once they have those they should test for an effect of season on that percentage.

c. I notice from the methods that collections varied with respect to baits used (banana or yeast), versus aspiration versus netting. The authors should provide the associated details on the Suppl Table as well check for batch effects again. It would be a shame if collection strategy were a hidden confounder.

d. Finally a question: the authors examine the genital arch of collected males to try to exclude simulans from the pools that way?

2) The heavy filtering of SNPs here will lead to some interesting ascertainment biases. Not only are they filtering out rare and private SNPs but the authors are requiring that the SNPs are found in EVERY population. We should expect this to be quite a biased set of SNPs, likely in the direction of enriching for balanced polymorphisms. While this is what the authors want to find I guess, it means that any statements regarding the percentage of SNPs that are under seasonally varying selection (and thus temporally balanced) will be significant overestimates. The authors at the very least will need to add significant caveats to their interpretations, but I would encourage them to consider redoing the analysis on an unfiltered set of SNPs as well, even if that means that the authors have to restrict portions of the analysis to within populations.

3) Line 742-I'm concerned that the authors are using too simple a regression here. They are assuming that allele frequency in fall is independent of allele frequency of spring within a population, which of course it is not. At the very least I think the authors' should be doing a repeated measures regression, but I wonder if there are even better ways to do this statistical testing, perhaps using a method of regression appropriate for autocorrelated timeseries observations.

4) Line 821-this model is not appropriate as observations of allele frequencies among populations are correlated via shared population history. The authors need to account for that covariance structure in this regression. See for instance Coop et al. 2010

5) Line 841-among population comparisons of the "rank p-value Fisher's method" test (can the authors come up with a better name?) are concerning as the authors are using number of reads as the data. If there are differences in read depth between samples, and there are, won't this test will have different power among different populations?

6) Lines 309-314. This analysis is unconvincing and the result quite weak. I note that the authors in this one tiny section are now presenting Pearson's R instead of R^2^. The coefficient of determination is very low for each sample here. Thus I feel that the authors are over interpreting what they have done.

7) The "Flipped model" strikes me as troubling philosophically. The authors' report a negative relation, so to make it line up with observations from other populations they are flipping the season labels? The authors need to do quite a bit of work to justify this as anything other than p-hacking, and the authors in my opinion should remove analysis of the "flipped" set from the paper.

*Reviewer #2:*

Overall this is a very nice study that convincingly shows an impact of seasonality on allele frequencies across many *Drosophila* populations, suggesting temporally-variable balancing selection may impact a large proportion of the *Drosophila* genome.

Are the estimates of the number of SNPs and strength of selection consistent with the observed patterns and decay of Fst? That is, how well does Fst and the decay of Fst in your simulations match the observed data?

Given these same parameters it might be fun (but admittedly speculative) to estimate what proportion of sites in the genome are affected by "seasonal draft".

What does the estimate of the strength of selection and number of sites affected tell us about load? Are local *Drosophila* population sizes sufficient that this is plausible without leading to extinction?

Of course the flipped model does provide more convincing evidence of some sort of seasonal effect, but I think I need a bit more convincing that the flipped model is justified other than the fact that it makes everything fit the preconceived model better.

How important is identication of individual causal loci (Line 653)? If the trait really is highly polygenic and there is limited concordance in loci among populations, how likely is this to be succesful? This doesn't seem to me the natural conclusion from the work presented. It seems to me that careful quant gen studies of phenotypes, selection gradients, and how Vg or Va change among populations and over time might be of more interest.

I'm generally not a fan of GO analyses but recognize that many readers will ask for such tests; I thought the paragraph presenting these results was appropriately cautious.

The definitions of spring and fall confused me somewhat. Are they based on per-location information on sampling abundance of *Drosophila*? It seems like there must be good weather station data for most of these locations; could a definition based on known thermal tolerances of *Drosophila* could be used?

What proportion of the ~1M SNPs filtered because they were not found across all populations show evidence of seasonal allele frequency change? It seems like population-specific variation is an interesting area that was left unexplored? Does this also explain some of the difference between Bergland 2014 and the present study?

*Reviewer #3:*

This manuscript addresses a classical question with unique data set. I was looking forward to reading it, but was disappointed by the statistical analysis, which I found both baffling and inappropriate.

I think the core of the problem is clearly and explicitly specified models and questions. Instead, you use a series of ad hoc tests that quickly made me lose faith in your conclusions.

It starts at the beginning (l. 186), where you perform a "test" of allele frequency change without specifying what you are testing, and why. After flipping between the 3-4 copies of the file I needed to have open to simultaneously look at text, figures, supplementary figures, and methods (which are not well written, see, e.g., the meaningless equation on l. 742), I think you are simply testing whether the spring frequencies at each SNP in each population differ significantly between spring and autumn under a very naive model, which you reject genome-wide (Figure 2A). But rather than tossing out this model and going back to the drawing board, you base all the remaining analyses on complicated intersections of p-values (which are inherently irrelevant in this paper, since you are not trying to identify individual loci).

This seems very odd. The main question of the paper is whether allele frequencies change consistently across populations, and the natural way to test this is using some kind of permutation scheme using the allele frequencies differences. You have naturally paired data, which probably can be treated as independent replicated (although this requires an argument about migration).

Having established this, why not fit the whole data set into a generalized mixed linear model, without a priori assumptions about the distribution of the random changes in allele frequencies (which must reflect both estimation error and random drift), and with explicit terms for selection (as a function of local climate) and (possibly) migration.

This would be so much easier to interpret than the ad hoc analyses you carry out, and would also avoid possible biases due to arbitrary p-value cut-offs for SNPs.

Using such an approach, it should be possible to estimate what fraction of the genome changes in a non-random fashion. Less sure about the strength selection, as I think this only has meaning within an accurate demographic model (which you do not have).

To sum up, I don't think you are doing your data justice here.

[Editors’ note: further revisions were suggested prior to acceptance, as described below.]

Thank you for submitting your work entitled "Broad geographic sampling reveals predictable, pervasive, and strong seasonal adaptation in *Drosophila*" for consideration by *eLife*. Your article has been reviewed by 3 peer reviewers, including Magnus Nordborg as the Reviewing Editor and Reviewer #1, and the evaluation has been overseen by a Senior Editor.

Our decision has been reached after consultation between the reviewers. Based on these discussions and the individual reviews below, we regret to inform you that our decision remains the same as after the first round: you are welcome to resubmit, but dramatic changes are (still) necessary. We are sorry if we didn't manage to explain this the first time around.

Although we remain convinced that your study may contain an important and interesting result – strong selection related to season/climate on a wide geographic scale – your analyses largely obscure this. A much simpler (and shorter!) paper that focused on convincingly demonstrating this fact would be much better.

Other major problems include the "flipped" analysis, which is not only post-hoc, but serves to illustrate that you should use climate data directly since the seasons obviously aren't really seasons, and the selection simulations which are based on models that are too simplistic.

Finally, arguing that the "haphazard" sampling is a strength, is, well, not convincing.

Our recommendation remains the same: given that the sampling is haphazard, simplify both claims and analysis in order to make the former rock solid. As you know, a recent pre-print by Buffalo and Coop argues that there is no signal in your first paper: here is your chance to prove that you were right after all.

*Reviewer #1:*

This version is much better than the previous one, but I still find the analysis confused and confusing. I will comment further below, but before getting to this, I should state clearly that I know how difficult this is. We are currently struggling with analogous data from Arabidopsis, and it clear that we lack an established framework for thinking about them. This is *not* textbook stuff.

A related, high-level comment is how ridiculously primitive we are from the point of view of basic ecology. We have no idea how selection is working, either in time or space, much less what the main selective pressures are. You write in your introduction that you "elected to solve these two related problems by (i) working as a consortium of *Drosophila* biologists (DrosRTEC or the *Drosophila* Real Time Evolution Consortium) and (ii) sampling in a somewhat haphazard manner." The first point is good, but the second is nonsense. The solution is surely not the kind of haphazard sampling described in this paper, but rather dense and non-random sampling in both time and space. (This is needed for Arabidopsis as well, btw)

When I first read this, I didn't even realize that all your samples were not collected in a single year. This raises the issue of why you don't take this into account. In our multi-year Arabidopsis field experiments, we find that there is often greater variability between years than between sites located near 1000 km from each other in a north to south direction (moving from deciduous forest to taiga).

First, what is the temporal correlation? The effect of space is clear from you PCA, but does time explain anything? How far do these flies move?

Second, you don't use climate data except to carry out your post hoc "flipping" of fall/spring labels. Did you try using climate data to explain what you see instead of a priori notions of spring and fall that you yourself dismiss as inadequate? We found that PCAs of multi-dimensional climate data was more strongly correlated with fitness than geographic proxies.

Another major unknown is the demographics of fruit flies. Your analyses assume (explicitly when you simulate selection) that each sampling location is a population (that dreaded population genetics abstraction that we all know doesn't really exist). You convincingly demonstrate that what you see cannot be due migration of randomly differentiated flies (because there is some kind concordance in changes globally), but this does not preclude that most of the allele frequencies be due to migration of selectively differentiated flies. Perhaps what happens is that there is selective die-off each winter, leading to cold-hardy survivors being overrepresented each spring – until they are swamped by southern migrants with vastly greater population sizes? Still selection, still adaptation, but with rather different predictions for the dynamics of allele frequencies. Talk about how fast populations adapt is less meaningful when we don't know what a population is. Returning to my initial point, to understand adaptation, we must understand the spatial and temporal scales over which allele frequencies chance. You don't have the sampling density to do this.

Glass half full: if you really see consistent changes despite all this, there must be some BIG signals out there…

I am convinced that you do see such changes, but my main comment remains that I think you could learn more from these data if you took several steps back and thought carefully about alternative models, and the simplest possible way to test. I suggested using the paired structure (you still need to show that pairs can be treated as independent, btw), but I think you can take it further than you have in estimating effects at individual SNPs. But it is not my job as reviewer to tell you how I would approach this, but rather to check whether I think your claims hold.

Point-by-point then, I am convinced by your permutations that there are large coordinated changes, but you still need to discuss the possible role of spatial and temporal correlations between pairs. Also think this point could be made better by going directly to estimates of allele frequency change, or beta from regression rather than involving p-values.

I find the whole "predictability" analysis convoluted and unconvincing. An attempt to shoehorn observations into an a priori framework. The leave-one-out analysis also requires some discussion of whether these points are independent, and I think it would not be necessary if you directly estimated consistent effects using permutation.

The comparison with clinal data is interesting, but again suffers from too strong priors of what is going on. You have climate data – why not simply check whether there are variables that explain both spatial and temporal shifts?

I find your analysis of the strength of selection across the genome unconvincing for the reasons outlined above: you have absolutely no idea of the demographic model, and treating each population as a close system (i.e. generating fall by sampling from spring) is simply not warranted.

The analysis of which SNPs are under selection is not convincing. The complete lack of correspondence between the original and the "flipped" model strongly suggests that the peaks are not to be trusted. Here is where jack-knifing and bootstrapping might be useful: to assess the robustness of your p-value estimates. My guess is that while your data is strong enough to show that there is selection in aggregate, getting down to individual loci is not possible. This also militates against relying of high-significance filters rather than the whole distribution of effect sizes.

*Reviewer #2:*

I have read the revised submission and the responses to review carefully. I still have major issues with the analysis done in this manuscript.

1) The authors still provide textbook case of p-hacking through their "flipped" analysis. Their justification-that the leave-one-out analysis tipped them off to a flipped effect-is not a justification at all. This entire section of the paper needs to be removed.

2) The binomial GLM that the authors are doing is inappropriate, as I pointed out in the initial submission. The "paired-fashion" of sampling helps nothing-the authors are simply testing against the null of their being no-difference in frequency between seasons assuming a binomial error model. The binomial error model here is inappropriate because spring and fall are not independent draws from the same parameterized distribution-allele frequency change is expected to occur do to drift. The authors are not accounting for this and it is a major flaw. I would suggest at the very least that the authors using a Nicholson et al. type framework for accounting properly for allele frequency change between seasons.

3) The justification that the authors give about "haphazard sampling" is risible. Adding noise does not add "inferential power" as the authors claim on line 174

4) The issue of biased ascertainment of SNPs has not been dealt with. The authors simply give their same estimates of genome-wide numbers of SNPs affected by seasonal selection and then follow it with a caveat. All such genome-wide estimates should be removed-you can't estimate them given your ascertainment conditions. Moreover you say on line 235 "Whether this SNP selection process generates bias in our estimates […] remains to be determined." This is unacceptable- your ascertainment definitely creates bias- this language brushes that under the rug.

5) The code for the permutation routines and the control SNP matching needs to be shared.

6) Lines 293-295 The authors are reporting that permution suggests the p-value of enrichment > 0.1. This suggests chance and nothing else is responsible for the observed effect despite the authors' conclusion of "robust evidence that parallele seasonal adaptation is a general feature…"

7) Line 309-no enrichment test has been described.

8) Line 330-Citing Gould here is silly. Pick a more appropriate citation.

9) Supplemental Figure 7-in the Bayesenv analysis that I asked for the "All" curve looks very different and goes against the conclusion that the authors are making. No explanation is given.

10) The ABC analysis, now described, is not using proper population genetic simulations of allele frequency change per generation due to drift + selection. As written there is only one generation of drift. This needs to be changed to take this analysis even moderately seriously. Moreover the code for these simulations needs to be shared.

11) Line 1020-In the corcordance regression the authors are doing a binned regression-this is never appropriate and the authors need to redo this analysis without binning.

*Reviewer #3:*

Overall this is an improved manuscript. Easier to read and follow, and better explained. There are several points I think that should still be addressed.

I am still not a fan of the flipped model. I agree that some of the evidence (predicting into the validation set, etc.) does indeed argue it's a better fit, but it still feels like ad-hoc subjective tweaking of the data until it fits well. I would prefer it to be removed from the paper -- I think show the original model and point out that some population show the reverse pattern and that matches with temperature. Perhaps even include the flipped model in the supplement. I would find that more convincing than the flipped model I think. In either case, the paragraph starting on line 445 should go, as even the authors admit this doesn't really show anything meaningful. The flipped model should also be removed from figures 2A and 2B as again it will by definition show a more convincing signal here.

As an alternative to the flipped model or presenting the data with the course labels of spring and fall, why not actually model the temperature data available? It would seem an objective a priori approach that should allow for differences in the flipped populations (i.e. presumably the difference in temperature the 3 weeks prior between Fall and Spring behaves differently for the flipped populations). Perhaps use mean temperature in the 3 weeks prior and/or the slope of the change of temperature over that time? I'm sure there are more creative/intelligent options, but I don't quite understand why the authors can't use this data instead of grossly categorizing things as spring or fall. I didn't see a good reason for not doing so in the response?

I find the authors treatment of enrichment odd. In some places it is presented as convincing evidence, and in others (line 563) it is disregarded because of absolute numbers. The logic on line 563 is fine of course, but I would like to see enrichment treated the same way throughout. On line 477 it is convincing as a log odds score, and in the paragraph starting on line 318 a modest percentage enrichment is considered good evidence.

I'd like to see a bit more exploration of the clustering. Figure 5D (visually) and the 100kb window analysis seem to suggest that clustering is on a relatively large scale, yet the analysis presented on 629 for % genome and s only investigates 5kb windows. If I'm understanding the ABC correctly it should be pretty fast to run, and it seems like running it on 50, 100, or even 500kb scale might be of interest. (To my eye some of the figures in S10 start to suggest a flattening of the ridge when done at 5kb scale). Certainly the data do appear to argue for a polygenic architecture, but whether this is ~50 windows or 5% of the genome I think isn't well differentiated.

Line 800: I agree with this logic about temperature and why some populations behave differently. I would have liked to see this prediction about temperature earlier in the introduction. Naively my first impression was that Fall populations would be adapted to cooler conditions and Spring to warmer. I see now why that is wrong, but I think stating up front that Fall populations are expected to reflect adaptation to warm summers would help some readers.

[Editors’ note: further revisions were suggested prior to acceptance, as described below.]

Thank you for resubmitting your work entitled "Broad geographic sampling reveals predictable, pervasive, and strong seasonal adaptation in *Drosophila*" for further consideration by *eLife*. Your revised article has been evaluated by Patricia Wittkopp as the Senior Editor and Magnus Nordborg as the Reviewing Editor.

Almost there! As you will see from the comments below, we got a fresh 3rd reviewer, who picked up on something we agree should be addressed, namely the likely role of inversions. We have two suggestions for how to proceed. Either: a) go through the manuscript and make sure you emphasize that much of the signal may be driven by inversions, and that it is impossible to know how polygenic this really is, or; b) provide additional analyses (e.g., of the X chromosome), to demonstrate that there is a signal independent of inversions.

*Reviewer #1:*

The authors were trying to confirm preliminary results suggesting that environmental changes accompanying seasonal change drive genome-wide allele frequency changes in *Drosophila*. This would give new insight into how selection works, and what factors might maintain genetic variation – at least in short-lived organisms. Although the detailed mechanisms are obscure, the authors use parallel changes over large geographic distances to argue convincingly that some form of seasonal selection must be taking place.

This is the third time I see this manuscript, so I will say no more than that it is greatly improved. I'm happy with it.

*Reviewer #2:*

This is a much improved, clearer version of the manuscript. The analyses are simpler and better explained, and the results I think come out clearer as a result.

*Reviewer #3:*

The strongest seasonal signal comes from inversions. If inversions are responding to seasonal selection, it is not surprising that the authors find parallel SNP changes across populations as the same inversions are shared globally. Unless the authors refocus the manuscript on parallel selection on inversions, their analyses need to be modified: almost all analyses use the full SNP set, but to study real parallel selection responses on the SNP level, the authors need to restrict their analysis on SNPs, which are not affected by inversions. To this end, it is important to keep in mind that inversions may suppress recombination also outside of the inversion, which makes it a bit challenging to determine the autosomal fraction that is not affected by inversions. A much better strategy would be to analyze the X-chromosome, which is the only major chromosome free of inversions. Unfortunately, the authors excluded this chromosome from their analyses.

Anyway, inspection of Figure 2D shows that the signal for seasonal SNPs is erased for regions outside of the inversions. Furthermore, a significant concordance pattern between seasonal and clinal SNPs outside of the inversion is restricted to 2L and 3R, the chromosomes with the strongest inversion effects. This could be interpreted as an effect of inversions on the genomic regions flanking the inversion.

How do the authors interpret a (presumably significant) underrepresentation of concordance SNPs on 3L?

Apart from my doubts about the significance of the seasonal selection signal, I would like to come back to the novel aspect of the manuscript-sharing seasonal SNPs across populations. The authors highlight, probably correctly, that seasonal adaptation is polygenic. This raises the question of whether parallel selection signatures are expected in differentiated populations. In my opinion two lines of reasoning speak against it: 1) probably more variants are segregating in the populations than required for seasonal adaptation (redundancy) 2) the frequencies of the seasonal SNPs most likely differ between the populations. Hence, SNPs closer to 50% are expected to respond more to the same selection pressure than SNPs with more extreme allele frequencies. This will lead to different power to detect the same selection response in differentiated populations.

Analyze the X-chromosome.

Remove the second season from the locations where two spring-fall pairs were included-only this makes the comparison unbiased.

Evaluate whether the spring-fall permutations remove the statistical issues of the GLM and Fisher's exact tests mentioned by the previous reviewers. Clarify that the matched controls were done on a sample basis, rather than across samples.

Clarify that the effective coverage was calculated per SNP.

The authors cite theoretical work, which suggests that seasonal SNPs may be maintained for highly restricted conditions (changing dominance)-do they find empirical support that these conditions are met in their data?

The significance of the manuscript to a broader audience could be increased by:

– A statement that the seasonal selection response is restricted to inversions-but I doubt that this is the message the authors would like to portray.

– A general discussion about the expectations of parallel selection signatures on the SNP level across populations and why the authors expect to see it (or find it against the expectations).

---

## [Author Response]

[Editors’ note: the authors resubmitted a revised version of the paper for consideration. What follows is the authors’ response to the first round of review.]

Reviewer #1:[…] 1) I have a number of concerns about the sampling design of this study:a. First and foremost there is no consistency to the sampling date between years or localities. For instance the Fall samples from Linvilla, PA were collected either in October or November. Indeed a quick search for average temps reveals that at their collection site the average daily temperature differs by ~ 10 degrees F between those two dates. Thus I'm concerned that "season" is not a properly replicated factor across the design and instead there could be significant batch effects.

The reviewer brings up an important point about the nature of seasons. A discussion of what defines a “season”, as it relates to the evolutionary dynamics of flies, is a major thread throughout the manuscript. In this revision, we have strengthened our treatment of this central topic throughout the Introduction, Results, and Discussion. Here, we list the text that we have added to the manuscript:

1. In the introduction (P4, L139-151), we highlight two challenges in studying seasonal evolution in multiple fly populations:

“Examining patterns of allele frequency change on seasonal timescales across a species’ range faces two problems. […] While it is possible, albeit logistically difficult, to sample flies according to some matching of seasons using the calendar window (e.g. first week of June for the “spring” and the first week of November for the “fall”) or using a pre-determined physiological time (e.g., “spring“ is the time after 21 cumulative growing degree days), it is not at all clear that any such matching would be meaningful in terms of the ecology, physiology, and evolution of local populations.”

2. With this statement of the challenges of defining seasons and sampling across them in multiple populations articulated, we go on to discuss the role of consortia based sampling (P4, L153-162) and then describe in detail how haphazard sampling is both logistically necessary and conceptually important (P4, L164-167):

“We chose to carry out haphazard sampling for logistical and inferential reasons. In this study, the “spring” sample was generally taken close to the time when the populations of *D. melanogaster* become locally abundant. […] Thus, a haphazard approach might in fact have the greatest inferential power provided that the signal of seasonal adaptation can be detected in such a complex dataset*”*

We recognize that the reviewer is bringing up an important point about proper replication, however we also hope that the reviewer understands that we are sampling flies from natural habitats and that short-term changes in environmental factors will always exist. To reiterate the argument that we advance in the manuscript, the semi-predictable changes in selection pressure observed in this system provides an ideal opportunity to decouple, and therefore identify, the environmental factors acting as strong selective agents in the wild.

b. I'm concerned about the possibility of contamination from accidental collection of D. simulans that should vary systematically between the seasons (we know that the abundance of sim vs mel in local collections changes dramatically throughout the season. While the authors are rightly trying to account for this by competitive mapping more assurances should be given that this is working. I would like to see a simple simulation demonstrating the power of the competitive mapping procedure-a straightforward way to do this would be to make synthetic pools of reads from melanogaster and simulans, while varying the percentage of simulans in the pool, and then take those simulated pools through the authors' mapping procedure. In addition, the authors should provide the percentage of reads that mapped to the sim assembly for each sample in Suppl Table 1. Once they have those they should test for an effect of season on that percentage.

The reviewer correctly points out that accidental collection of *D. simulans* can occur, and if the probability of such contamination varies between seasons this may influence our inferences of seasonal adaptation. We have taken the following steps to address the reviewer’s concerns:

1. We conducted simulations showing that we can accurately estimate the fraction of *D. simulans* reads through competitive mapping. This work is now described in the Materials and methods (page 23, L912-926) and Supplemental Figure 12:

“In silico simulation analysis shows that competitive mapping accurately estimates the fraction of *D. simulans* contamination. […] The level of residual *D. simulans* contamination does not correlate with contamination rate and is roughly 9% (Supplemental Figure 13B).”

2. We show that the local density of seasonal polymorphisms is not correlated with the rate of *D. simulans* reads mapping to the *D. melanogaster* genome even with competitive mapping (page 23, L926-928):

“Additionally, the proportion of seasonal sites (top 1%) is not correlated with the rate of residual mapping of *D. simulans* reads to the *D. melanogaster* genome as inferred through our simulation (r = -0.0026, p=0.93)”.

3. We show that there is no bias in the percent of *D. simulans* contamination between our spring and fall samples (t-test p=0.90; added to P23, L907-910):

“For the populations used in this analysis, the average proportion of reads mapping preferentially to D. simulans was less than 1% (see Supplemental Table 1), and there was no significant difference in the level of D. simulans contamination between spring and fall samples (t-test p=0.90).”

4. We separated *D. melanogaster* from *D. simulans* males by examination of the genital arch, and only sequenced those individuals likely to be *D. melanogaster*. Residual *D. simulans* contamination is likely due to human error in this process. We state clearly that we use the gential arch on P22 L876:

“Flies were confirmed to be *D. melanogaster* by examination of the male genital arch.”

5. As in the previously submitted manuscript, we excluded any samples with high *D. simulans*. We state this in the Materials and methods. Intriguingly, both of the excluded samples were from the fall when *D. simulans* was generally thought to be in low abundance (Schmidt 2011). We interpret the high *D. simulans* contamination rate from fall samples to be a consequence of poorly tuned priors on the part of the person doing the sorting. We have decided to leave this interpretation out of the manuscript, however.

c. I notice from the methods that collections varied with respect to baits used (banana or yeast), versus aspiration versus netting. The authors should provide the associated details on the Suppl Table as well check for batch effects again. It would be a shame if collection strategy were a hidden confounder.

We have added collection substrate, method, and location type (residential, orchard, etc.) to Supplementary Table 1.

We also tested whether collection locale (orchard vs. residential), collection method (aspirator, net, trap), collection substrate (apple, banana, compost/pomace pile, windfall fruit, other [cherries, peaches, mixed fruit]) in the spring or fall collection are correlated with genome-wide predictability score. Using a simple ANOVA, the smallest p-value across all tests is 0.45. We have added this analysis to the main text where we test whether predictability scores are correlated with other environmental values (P10-11, L401-407):

“However, genome-wide predictability scores are not correlated with spring or fall collection date (p = 0.6 and 0.9, respectively), cumulative growing degrees (p = 0.5, 0.86), collection locale (residential vs. orchard, p = 0.6, 0.4), collection method (aspirator vs. net vs. trap, p = 0.9, 0.9), collection substrate (contrasting various types of fruit, compost, p = 0.8, 0.9), percent D. simulans contamination (0.8, 0.8), latitude (p = 0.9), or the difference in cumulative growing degree days between spring and fall (p = 0.46)”

We note that there may be biologically meaningful interactions between the aspects of the environment mentioned above which could contribute to the observed signals of seasonal adaptation. And, we completely agree with the reviewer that this is an important point. We hope that once the sample size of the seasonal dataset goes up to hundreds or thousands such analyses would become powerful enough to identify the effects of the myriad selective forces underlying rapid adaptation in the wild.

d. Finally a question: the authors examine the genital arch of collected males to try to exclude simulans from the pools that way?

Yes, we did. Please see above.

2) The heavy filtering of SNPs here will lead to some interesting ascertainment biases. Not only are they filtering out rare and private SNPs but the authors are requiring that the SNPs are found in EVERY population. We should expect this to be quite a biased set of SNPs, likely in the direction of enriching for balanced polymorphisms. While this is what the authors want to find I guess, it means that any statements regarding the percentage of SNPs that are under seasonally varying selection (and thus temporally balanced) will be significant overestimates. The authors at the very least will need to add significant caveats to their interpretations, but I would encourage them to consider redoing the analysis on an unfiltered set of SNPs as well, even if that means that the authors have to restrict portions of the analysis to within populations.

We agree with the reviewer and thank them for bringing up this interesting and important point. We now discuss the limitations of our analysis, specifically with respect to ascertainment bias caused by minor allele frequency cut-off and the requirement for polymorphism in all populations. We have now addressed this point at several places in the manuscript and have performed additional analyses. We outline these changes here:

1. We acknowledge the general problem in the beginning of the results where we write on P6 L234-237:

“Whether this SNP selection process generates bias in our estimates of the strength and magnitude of seasonally variable selection remains to be determined but note that the smaller set still represents approximately half of all detected SNPs.”

2. Again, on P15 L610-615, where we write:

“We note that our analysis is potentially biased because we are conditioning on polymorphisms that are present in all populations tested, including North American and European ones with different long-term dynamics and histories of colonization, expansion, and admixture (Keller 2007; Bergland et al. 2016; Kapopoulou et al. 2018). The filtering requirement that we impose may therefore bias our SNP set to those which are subject to balancing selection, causing an upward bias in our estimates.”

3. We discuss our how estimates of the strength and extent of seasonal allele frequency changes are qualitatively similar when we include the broader set of 1.75M SNPs (P16 L615):

“We note, however, there are no qualitative differences found as a function of filtering for common SNPs (compare Figure 4B to Supplemental Figure 9).”

The reviewer also raises the interesting suggestion of performing analyses of seasonal adaptation within populations. On page P20 L 771-765 we discuss how the dataset that we have assembled only has high power for analysis across populations:

“Note that we generally lack power to detect alleles that cycle in a small subset of populations or are private to a single population. […] Thus, our conclusions are limited to the shared genetic basis of seasonal adaptation as revealed by the cycling of common SNPs*.”*

3) Line 742 – I'm concerned that the authors are using too simple a regression here. They are assuming that allele frequency in fall is independent of allele frequency of spring within a population, which of course it is not. At the very least I think the authors' should be doing a repeated measures regression, but I wonder if there are even better ways to do this statistical testing, perhaps using a method of regression appropriate for autocorrelated timeseries observations.

We thank the reviewer for their helpful suggestion. However, we believe that our model is appropriate because we are treating the data in a paired fashion.

4) Line 821 – this model is not appropriate as observations of allele frequencies among populations are correlated via shared population history. The authors need to account for that covariance structure in this regression. See for instance Coop et al. 2010

We have taken the reviewer’s recommendation and performed an additional analysis using the recommended bayenv regression. This new analysis does not substantially change the results or any of the conclusions. We have added this comparison to the manuscript (P13, L521-525; Supplementary Figure 7):

“Seasonally varying SNPs were identified among these 18 populations using both the original and flipped seasonal labels. […] We find that the rate of parallelism increases with increasingly stringent seasonal and clinal significance thresholds (Figure 3A) using either model of clinality (Supplemental Figure 7).”

5) Line 841 – among population comparisons of the "rank p-value Fisher's method" test (can the authors come up with a better name?) are concerning as the authors are using number of reads as the data. If there are differences in read depth between samples, and there are, won't this test will have different power among different populations?

The ranking of the Fisher’s exact test statistic (which produces the rank p-values that are the input of the Fisher’s method test) is performed within each population. The power is relative to the number of observations (SNPs) being ranked. Within each population, ranking occurs among SNPs with similar allele frequencies and depths (see methods for binning). The number of bins is the same for each population, and each population has equivalent power. We have clarified this in the Materials and methods, P27 L1096-1098:

“However, as our rank p-value Fisher’s method relies only on the rank within a paired sample, and the consistency of the rank across paired samples, there is not likely to be an artificially inflated seasonal signal due to incorrectly estimated sampling error”.

We use the acronym, “RFM” throughout the manuscript after defining it once.

6) Lines 309-314. This analysis is unconvincing and the result quite weak. I note that the authors in this one tiny section are now presenting Pearson's R instead of R^2^. The coefficient of determination is very low for each sample here. Thus I feel that the authors are over interpreting what they have done.

We appreciate the reviewer’s prudent interpretation of this result. It reflects our own cautious interpretation that we included in the original manuscript and remains in the current one on P11, L424 (“While this correlation of three points is not significantly different from zero”.)

We also note that the use of R instead of R^2^ is appropriate here because we care about the sign of the correlation between the model based and observed genome-wide predictability scores and not claiming that our model explains a substantial part of the total variance.

We also wish to remind the reviewer that the analysis in question is part of a larger analysis examining the validity of our thermal limit model which led to the development of the ‘flipped’ model. The ‘flipped model’ of seasonal allele frequency change has a substantially stronger signal when assessed using various, orthogonal tests that we present in the paper.

7) The "Flipped model" strikes me as troubling philosophically. The authors' report a negative relation, so to make it line up with observations from other populations they are flipping the season labels? The authors need to do quite a bit of work to justify this as anything other than p-hacking, and the authors in my opinion should remove analysis of the "flipped" set from the paper.

We respectfully disagree with the reviewer that we are engaged in p-hacking. Indeed, the flipped model arose naturally out of our leave-one analysis that we perform. In that analysis, we observed a prominent signal that several populations display strong allele frequency changes in the “opposite” direction from the rest of the paired samples. That result, we believe, is unambiguous.

We would like to note that we incorrectly stated that this leave-one out analysis represents a validation of the overall results. Instead we use it – and now describe it as such – to test whether the signal comes from just a few populations or from the sample as a whole and whether individual populations conform to that signal. We found that most do but some do not and show as strong but reverse, “flipped” pattern.

We then further show that biologically plausible thermal limits are correlated with this flipping phenomenon. Of course, another plausible explanation is that the season labels were mistakenly swapped at some point in sample handling or library preparation. While possible, we believe this is unlikely as there is no reason for why such a mistake would take place in a way that relates to the thermal model.

Regardless of the cause of the flipping, it is clear that the flipped model offers substantial improvement in the signal of seasonal evolution, even for tests which are orthogonal (e.g., clinal concordance and ability to predict an independent dataset from Bergland 2014 PloS Genetics paper). Note that we now discuss this later possibility in the manuscript where we write on P11, L408-411:

“Although we cannot rule out other explanations such as inadvertent sample swapping, our analysis is generally consistent with a model which suggests that changes in selection pressure within and among seasons may lead to dramatic changes in allele frequencies.“

Reviewer #2:[…] Are the estimates of the number of SNPs and strength of selection consistent with the observed patterns and decay of Fst? That is, how well does Fst and the decay of Fst in your simulations match the observed data?Given these same parameters it might be fun (but admittedly speculative) to estimate what proportion of sites in the genome are affected by "seasonal draft".

This is an interesting question and suggestion but at the moment is outside the scope as the simulations were conducted using unlinked loci in order to provide us with a sense of statistical power and the number of seasonal sites and the range of fluctuations consistent with our data. The more realistic model that would include linkage are much more challenging to carry out and the ability to fit the data to the model is limited by the nature of pooled datasets. We hope to carry out such an analysis in the future with the data that includes haplotype information.

What does the estimate of the strength of selection and number of sites affected tell us about load? Are local *Drosophila* population sizes sufficient that this is plausible without leading to extinction?

We refer the reviewer to Wittmann et al. PNAS 2017, where we discuss how to interpret the plausibility of strong, multilocus seasonal evolution in the context of load. We also refer the reader to Bergland et al. PloS Genetics 2014 where we discuss various plausibility arguments. Importantly, as we now discuss in the Results and Discussion, we do not claim that the detected sites are all causal and in fact believe that the majority must be cycling due to linked selection. The answer to the reviewer’s very good and appropriate question will need to wait until we and others arrive at a set of likely causal loci – such a fine mapping will require much larger datasets, and additional studies that complement sequencing, that we hope to collect in the future.

Of course the flipped model does provide more convincing evidence of some sort of seasonal effect, but I think I need a bit more convincing that the flipped model is justified other than the fact that it makes everything fit the preconceived model better.

The reviewer brings up an interesting point about the flipped model. We wish to highlight the following points:

1. The leave-one-out analysis which motivated the flipped model presents a striking feature of the data – that some populations have strong allele frequency fluctuations at many loci that go in the opposite direction from the remaining samples.

2. We provide evidence that the flipped model provides stronger signals of seasonal adaptation in analyses that are orthogonal to the flipping, i.e., the ValidationSet analysis, the clinal analysis, and enrichment with seasonal SNPs identified by Bergland et al. 2014.

3. The pre-conceived model, of course, is that spring *is* spring. The flipped model provides support for an alternative model in which local weather conditions provide a better predictor of adaptive dynamics over the growing season of flies than our limited definition of seasons.

How important is identification of individual causal loci (Line 653)?

We agree that the identification of the causal loci is not of core importance in this study. Ultimately, the identification of individual causal loci will be important because it will allow us to understand how strong selection on highly polygenic quantitative traits is realized at the genomic level. That being said we do succeed in identification of some regions and in some cases (such as for Tep 2,3) we have prior data suggestive of functional importance. We do believe that this analysis is important to do just to emphasize that the selection is polygenic and much additional data will be required to really identify individual loci.

If the trait really is highly polygenic and there is limited concordance in loci among populations, how likely is this to be successful?

It is unclear how successful this endeavor will be, especially given the current limitations of association mapping in *Drosophila*. This is an excellent question but cannot be addressed in the current study.

This doesn't seem to me the natural conclusion from the work presented. It seems to me that careful quant gen studies of phenotypes, selection gradients, and how Vg or Va change among populations and over time might be of more interest.

We agree and thank the reviewer for the interesting suggestion. This work is currently outside the scope of the present study but ultimately full understanding of the fluctuating selection will require such an integration.

I'm generally not a fan of GO analyses but recognize that many readers will ask for such tests; I thought the paragraph presenting these results was appropriately cautious.

Thank you! We are not fond of GO analyses for such types of data ourselves and the reviewer is correct that many people are curious about the results of this test. Therefore, we have worded this sentence cautiously.

The definitions of spring and fall confused me somewhat. Are they based on per-location information on sampling abundance of *Drosophila*? It seems like there must be good weather station data for most of these locations; could a definition based on known thermal tolerances of *Drosophila* could be used?

The reviewer makes and interesting point and we would refer them to discussion of this point during numerous passages throughout the manuscript. See our response to Reviewer 1’s first point.

What proportion of the ~1M SNPs filtered because they were not found across all populations show evidence of seasonal allele frequency change? It seems like population-specific variation is an interesting area that was left unexplored? Does this also explain some of the difference between Bergland 2014 and the present study?

This is an extremely interesting question, which unfortunately we do not have much power to explore. Identifying seasonal SNPs in single populations is challenging with only end-point data because of limitations in power. Note that we do not believe that the differences between this work and Bergland et al. 2014 can be explained by the use of private SNPs as the work by Bergland et al. 2014 relied on heavier filtering, with MAF across all populations (3 years of PA plus the clinal samples) > 0.15.

Reviewer #3:This manuscript addresses a classical question with unique data set. I was looking forward to reading it, but was disappointed by the statistical analysis, which I found both baffling and inappropriate.I think the core of the problem is clearly and explicitly specified models and questions. Instead, you use a series of ad hoc tests that quickly made me lose faith in your conclusions.It starts at the beginning (l. 186), where you perform a "test" of allele frequency change without specifying what you are testing, and why. After flipping between the 3-4 copies of the file I needed to have open to simultaneously look at text, figures, supplementary figures, and methods (which are not well written, see, e.g., the meaningless equation on l. 742), I think you are simply testing whether the spring frequencies at each SNP in each population differ significantly between spring and autumn under a very naive model, which you reject genome-wide (Figure 2A). But rather than tossing out this model and going back to the drawing board, you base all the remaining analyses on complicated intersections of p-values (which are inherently irrelevant in this paper, since you are not trying to identify individual loci).

We thank the reviewer for their comments and helpful suggests for clarifying our manuscript. We have taken these comments to heart and have thoroughly cleaned up the analyses and simplified and streamlined their presentation. We have modified the Materials and methods to make it more comprehensive, and have also substantially re-written the Introduction, Results, and Discussion. We also moved the figures into the main body of the text to aid in in the review process.

This seems very odd. The main question of the paper is whether allele frequencies change consistently across populations, and the natural way to test this is using some kind of permutation scheme using the allele frequencies differences. You have naturally paired data, which probably can be treated as independent replicated (although this requires an argument about migration).

We agree and in fact this is exactly what we do. We now make this key point much more clearly and we thank the reviewer for pointing out that this key part of the analysis was obscure in the previous version.

Having established this, why not fit the whole data set into a generalized mixed linear model, without a priori assumptions about the distribution of the random changes in allele frequencies (which must reflect both estimation error and random drift), and with explicit terms for selection (as a function of local climate) and (possibly) migration.

We look forward to implementing this suggestion in future analyses which contain more paired spring-fall samples. Further, we believe that non-parametric models such as Fisher’s exact test rank p-value method that we implement here coupled with permutations to arrive at the null distribution is likely to be the safest approach to take. In our experience it is almost impossible to accurately model error in these datasets and one generally ends up with inflated p-values. In any case, while this is an excellent suggestion, we feel that it is outside the scope of the present manuscript.

This would be so much easier to interpret than the ad hoc analyses you carry out, and would also avoid possible biases due to arbitrary p-value cut-offs for SNPs.

We would like to point out that we generally present results across a range of p-value thresholds and ask whether the signal of generally orthogonal tests get stronger as one goes to more and more stringent cutoffs. This is a general practice in such studies and we believe it is a good practice.

Using such an approach, it should be possible to estimate what fraction of the genome changes in a non-random fashion. Less sure about the strength selection, as I think this only has meaning within an accurate demographic model (which you do not have).

Please note that our approach to estimating the fraction of the genome that is subject to seasonal fluctuations is not based on a subset of significant SNPs – rather it is based on the genome-wide distribution of a semi-parametric test-statistic. We end up with a ridge of plausible values and report it as such.

To sum up, I don't think you are doing your data justice here.

We agree that our original presentation was too convoluted to do the data justice. We hope that the revised manuscript allays these concerns.

[Editors’ note: what follows is the authors’ response to the second round of review.]

Our decision has been reached after consultation between the reviewers. Based on these discussions and the individual reviews below, we regret to inform you that our decision remains the same as after the first round: you are welcome to resubmit, but dramatic changes are (still) necessary. We are sorry if we didn't manage to explain this the first time around.Although we remain convinced that your study may contain an important and interesting result – strong selection related to season/climate on a wide geographic scale – your analyses largely obscure this. A much simpler (and shorter!) paper that focused on convincingly demonstrating this fact would be much better.Other major problems include the "flipped" analysis, which is not only post-hoc, but serves to illustrate that you should use climate data directly since the seasons obviously aren't really seasons, and the selection simulations which are based on models that are too simplistic.Finally, arguing that the "haphazard" sampling is a strength, is, well, not convincing.Our recommendation remains the same: given that the sampling is haphazard, simplify both claims and analysis in order to make the former rock solid. As you know, a recent pre-print by Buffalo and Coop argues that there is no signal in your first paper: here is your chance to prove that you were right after all.

As you will see from the revised manuscript we have taken your comments to heart and significantly streamlined the paper. It is much shorter and simpler now and as you suggested is fully focused on detecting the signal of seasonal adaptation and defining the environmental factors that correlate best with the detected adaptation.

Briefly, as suggested by the reviewers we use several distinct statistical approaches to detect seasonal adaptation: (1) GLM, (2) a method based on Fisher’s exact test, and (3) Bayenv. Furthermore, we use the paired structure of our data to create appropriate null distributions via permutations of the Spring/Fall labels and retaining all other structure of the data. We show that all three approaches indeed detect a significant signal of seasonal adaptation above what we expect from permutations.

We show that the detected seasonal SNPs are significantly concordant with clinal patterns even when we detect seasonal adaptation using a set of populations in Europe and California and testing for concordance with the clinal patterns along the East Coast of North America. This avoids the possibility of shared seasonal migration patterns affecting the signal.

We have performed additional analysis studying the genomic distribution of seasonal SNPs. In contrast to the previous version of this manuscript, we now present evidence that seasonal SNPs are significantly enriched on chromosomes 2L and 3R, and are further enriched in regions associated with inversions on 2L, 3L, and 3R. We show that seasonal SNPs are significantly concordant with clinal patterns (the Fall variants tend to be more common in the North) and that this signal is found both inside and outside the inversions. The totality of the data supports the model of polygenic adaptation with a possibly more prominent role of the coadapted gene complexes associated with inversions. Although the nature of our dataset precludes us from an in-depth analysis of the identification of individual causal sites and the delineation of the causal role of specific inversions on seasonal adaptation, we believe that these results strengthen the claim that seasonal adaptation is an important feature of *Drosophila* populations living in temperate environments.

We finish the paper by building an environmental model that attempts to explain varying patterns of concordance of individual populations with the overall pattern of seasonal fluctuations using a leave-one-out analysis. We show that the best fit model uses the number of days in the weeks prior to spring sampling with maximum temperature above ~32°C and the number of days in the fall with minimum temperature below ~5°C. These values roughly correspond to upper and lower thermal limits for *D. melanogaster* physiology and the length of a single generation.

In order to streamline our message, we have removed any reference to the “flipped” model and have also removed our ABC analysis.

We believe the paper is much stronger, more succinct, and makes an important point about the speed of evolution and maintenance of genetic variation.

Reviewer #1:This version is much better than the previous one, but I still find the analysis confused and confusing. I will comment further below, but before getting to this, I should state clearly that I know how difficult this is. We are currently struggling with analogous data from Arabidopsis, and it clear that we lack an established framework for thinking about them. This is not textbook stuff.A related, high-level comment is how ridiculously primitive we are from the point of view of basic ecology. We have no idea how selection is working, either in time or space, much less what the main selective pressures are. You write in your introduction that you "elected to solve these two related problems by (i) working as a consortium of *Drosophila* biologists (DrosRTEC or the *Drosophila* Real Time Evolution Consortium) and (ii) sampling in a somewhat haphazard manner." The first point is good, but the second is nonsense. The solution is surely not the kind of haphazard sampling described in this paper, but rather dense and non-random sampling in both time and space. (This is needed for Arabidopsis as well, btw)

We agree that it is a hard problem and instead of arguing the point about the value of haphazard sampling we removed this claim from the paper.

When I first read this, I didn't even realize that all your samples were not collected in a single year. This raises the issue of why you don't take this into account. In our multi-year Arabidopsis field experiments, we find that there is often greater variability between years than between sites located near 1000 km from each other in a north to south direction (moving from deciduous forest to taiga).

The reviewer is correct that our dataset includes samples collected across multiple years. As we point out in the manuscript, we take the multi-year sampling into account in a number of ways:

1. In order to minimize bias caused by longer term sampling of some localities, we only use two years at most from any one sampling locality.

The 20 paired spring-fall samples were collected at 15 sampling localities.

Therefore 5 of the localities were sampled for multiple years.

2. Our GLM model includes a `population` term. In this model, we encode population as an unordered factor representing each of the 20 paired spring-fall samples. Thus, the GLM includes a “year” term, although it is embedded in a locality-by-year interaction (the population term).

3. The Bayenv model that we use would presumably capture any strong genetic differentiation between years in the genetic covariance matrix. Therefore, the signals of seasonal adaptation that we observe using the

Bayenv model should be robust to year-to-year variation.

We agree that year-to-year variation in the genetic structuring of *Drosophila* is likely an important aspect of the evolutionary dynamics of this species. We believe that the present manuscript will provide a good starting point for future investigation of this topic.

First, what is the temporal correlation? The effect of space is clear from you PCA, but does time explain anything? How far do these flies move?

The PCA is dominated by the spatial signal which is very robust and clearly persistent despite seasonal adaptation. We do not detect any PCs that do correlate with season and state so in the paper: “No PC is associated with season, as defined by the time of collection, following correction for multiple testing, implying that the spatial differentiation dominates patterns of SNP variation across these samples.” (line ~155)

Second, you don't use climate data except to carry out your post hoc "flipping" of fall/spring labels. Did you try using climate data to explain what you see instead of a priori notions of spring and fall that you yourself dismiss as inadequate? We found that PCAs of multi-dimensional climate data was more strongly correlated with fitness than geographic proxies.

We no longer do any flipping, and yes do build an environmental model that explains the parallelism of individual populations with the rest of the populations surprisingly well. This is a major component of the paper now and we have worked to clarify our approach in the manuscript.

We do indeed conclude that simple “spring” and “fall” labels are likely inadequate descriptors across a broad geographic range. However, we believe that it is a reasonable a priori starting point for analysis.

Another major unknown is the demographics of fruit flies. Your analyses assume (explicitly when you simulate selection) that each sampling location is a population (that dreaded population genetics abstraction that we all know doesn't really exist). You convincingly demonstrate that what you see cannot be due migration of randomly differentiated flies (because there is some kind concordance in changes globally), but this does not preclude that most of the allele frequencies be due to migration of selectively differentiated flies. Perhaps what happens is that there is selective die-off each winter, leading to cold-hardy survivors being overrepresented each spring – until they are swamped by southern migrants with vastly greater population sizes? Still selection, still adaptation, but with rather different predictions for the dynamics of allele frequencies. Talk about how fast populations adapt is less meaningful when we don't know what a population is. Returning to my initial point, to understand adaptation, we must understand the spatial and temporal scales over which allele frequencies chance. You don't have the sampling density to do this.

We completely agree that understanding the spatial and temporal scales of allele frequency change is important and that the interpretation of those data are dependent on the nature of the “population”. Factors such as extinction, migration, population growth, and deme-size (to name a few) will likely have a major impact on allele frequency change through time. While our dataset permits us the opportunity to study some aspects of these meta-population features, we are clearly limited in addressing many of these nuanced features. Indeed, we are sample limited. We now discuss what is known/unknown about the meta-population dynamics of *Drosophila* in the Discussion section and draw attention to this important problem, including the need for increased sample density.

Glass half full: if you really see consistent changes despite all this, there must be some BIG signals out there…

We agree.

I am convinced that you do see such changes, but my main comment remains that I think you could learn more from these data if you took several steps back and thought carefully about alternative models, and the simplest possible way to test. I suggested using the paired structure (you still need to show that pairs can be treated as independent, btw), but I think you can take it further than you have in estimating effects at individual SNPs. But it is not my job as reviewer to tell you how I would approach this, but rather to check whether I think your claims hold.

The goal of this manuscript is to provide evidence that seasonal adaptation is occurring in multiple populations across a portion of *D. melanogaster*’s range. We provide evidence of seasonal adaptation using three methods, one of which takes into account the genetic covariation between samples. We provide evidence that the seasonal SNPs that we observe are enriched with seasonal SNPs identified previously in Bergland et al. 2014, that patterns of seasonal adaptation are concordant across time and space, and that aspects of weather are correlated with the direction of allele frequency change. Taken together, we believe that our results go above and beyond the estimation of the effects of individual SNPs.

Point-by-point then, I am convinced by your permutations that there are large coordinated changes, but you still need to discuss the possible role of spatial and temporal correlations between pairs. Also think this point could be made better by going directly to estimates of allele frequency change, or beta from regression rather than involving p-values.

We now discuss the role of spatial and temporal correlations between pairs whenever we discuss the use of the Bayenv model.

We focus our analysis on the distribution of p-values, not betas, because p-values also include information on the precision of allele frequency change. In our experience, some sites with very large betas have noisy estimates of allele frequency (i.e., low read depth), or are at very low allele frequency. Selecting the sites with large betas (x-axis) would focus on a very different set of SNPs than focusing on sites with low p-values (y-axis). We agree that a study of the effect sizes of allele frequency change could be useful, however we would respectfully ask that this analysis wait for future analyses that incorporate more data and will allow for much greater resolution.

I find the whole "predictability" analysis convoluted and unconvincing. An attempt to shoehorn observations into an a priori framework. The leave-one-out analysis also requires some discussion of whether these points are independent, and I think it would not be necessary if you directly estimated consistent effects using permutation.

The “predictability” analysis that we applied grew out of a reasonable question that we asked as we acquired new seasonal samples: Are there predictable seasonal shifts in a new paired sample, compared to the previously acquired data? This work was generalized to the “leave-one-out” analysis for the data-set that we present in the manuscript. We agree that permutations are an important aspect of this analysis and we show that there are partially consistent effects associated with “season” using three different approaches (discussed above).

The comparison with clinal data is interesting, but again suffers from too strong priors of what is going on. You have climate data – why not simply check whether there are variables that explain both spatial and temporal shifts?

This is a completely reasonable question, and we suggest that this question is better answered in a separate study. Answering this question requires careful thought on what environmental factors to study, and how to meaningfully summarize environmental features recently felt by populations (e.g., weather as we discuss in our paper) and long-term environmental differences (e.g., climatic factors that consistently vary among populations).

As described above while we cannot make strong statements about exactly how temporal and spatial dynamics interact we do discuss multiple possibilities in the Discussion. Specifically, shared migration cannot generate the signal because we see concordance between seasonal and spatial patterns in very distantly located populations. Moreover, large scale seasonal migration is generally very unlikely given our data because populations retain their spatial identity on the spatial PCA year after year. (This is in contrast to the patterns seen in D. simulans where we do think that flies do not overwinter locally but re-migrate from more southern locations year after year and consequently lack much of the spatial structure (Machado et al. 2016)). Some more local patterns of extinction-recolonization that must involve selection in order for the seasonal patterns to be detectable are plausible but we cannot make any rigorous claims about them. We do discuss them however.

I find your analysis of the strength of selection across the genome unconvincing for the reasons outlined above: you have absolutely no idea of the demographic model, and treating each population as a close system (i.e. generating fall by sampling from spring) is simply not warranted.

We have removed the analysis of the strength and ubiquity of selection.

The analysis of which SNPs are under selection is not convincing. The complete lack of correspondence between the original and the "flipped" model strongly suggests that the peaks are not to be trusted. Here is where jack-knifing and bootstrapping might be useful: to assess the robustness of your p-value estimates. My guess is that while your data is strong enough to show that there is selection in aggregate, getting down to individual loci is not possible. This also militates against relying of high-significance filters rather than the whole distribution of effect sizes.

We removed this analysis and no longer attempt to detect individual loci under selection or to construct a flipped model. These questions will be better addressed with more dense sampling that we are carrying out in collaboration with the *D. melanogaster* research community.

Reviewer #2:I have read the revised submission and the responses to review carefully. I still have major issues with the analysis done in this manuscript.1) The authors still provide textbook case of p-hacking through their "flipped" analysis. Their justification-that the leave-one-out analysis tipped them off to a flipped effect-is not a justification at all. This entire section of the paper needs to be removed.

We respectfully disagree that this is p-hacking. Nonetheless we have removed this analysis from the paper. This made the paper much more streamlined and does not raise the issues of how such post hoc analyses ought to be done correctly.

2) The binomial GLM that the authors are doing is inappropriate, as I pointed out in the initial submission. The "paired-fashion" of sampling helps nothing-the authors are simply testing against the null of their being no-difference in frequency between seasons assuming a binomial error model. The binomial error model here is inappropriate because spring and fall are not independent draws from the same parameterized distribution-allele frequency change is expected to occur do to drift. The authors are not accounting for this and it is a major flaw. I would suggest at the very least that the authors using a Nicholson et al. type framework for accounting properly for allele frequency change between seasons.

We agree with the reviewer that the GLM model is not the perfect model to analyze pooled allele frequency change data. As we show in the supplement, for instance, the nominal p-values of the GLM are likely to be inflated. However, because we construct the null distribution using permutations that flip the Spring and Fall labels for each population we believe that our analysis is appropriate against this null. In the revised manuscript we use two alternative approaches – a Bayesian Bayenv method and the RFM method based on Fisher’s exact test – all of which detect similar strength of the seasonal signal against the permutation. We thus are fully confident that our analysis is robust.

3) The justification that the authors give about "haphazard sampling" is risible. Adding noise does not add "inferential power" as the authors claim on line 174

We removed this argument from the manuscript.

4) The issue of biased ascertainment of SNPs has not been dealt with. The authors simply give their same estimates of genome-wide numbers of SNPs affected by seasonal selection and then follow it with a caveat. All such genome-wide estimates should be removed-you can't estimate them given your ascertainment conditions. Moreover you say on line 235 "Whether this SNP selection process generates bias in our estimates […] remains to be determined." This is unacceptable- your ascertainment definitely creates bias- this language brushes that under the rug.

We have removed this analysis from the manuscript.

5) The code for the permutation routines and the control SNP matching needs to be shared.

Code for all analyses can be found at https://github.com/machadoheather/dmel_seasonal_RTEC.

The code to run the permutations can be found below.

GLM:

https://github.com/machadoheather/dmel_seasonal_RTEC/seasonal_glm/ RFM:

https://github.com/machadoheather/dmel_seasonal_RTEC/seasonal_rfm/

Bayenv:

https://github.com/machadoheather/dmel_seasonal_RTEC/seasonal_baye

The code for the control SNP matching for the latitudinal and seasonal concordance can be found here:

https://github.com/machadoheather/dmel_seasonal_RTEC/create_input_fil

6) Lines 293-295 The authors are reporting that permution suggests the p-value of enrichment > 0.1. This suggests chance and nothing else is responsible for the observed effect despite the authors' conclusion of "robust evidence that parallele seasonal adaptation is a general feature…"

We have changed this section entirely. We now report three different statistical methods and detect seasonal adaptation robustly.

7) Line 309-no enrichment test has been described.

This section has been deleted.

8) Line 330-Citing Gould here is silly. Pick a more appropriate citation.

This section has been deleted.

9) Supplemental Figure 7-in the Bayesenv analysis that I asked for the "All" curve looks very different and goes against the conclusion that the authors are making. No explanation is given.

The differences are present only at low p-values where there are few SNPs. We have updated the figure with error bars to reflect this.

10) The ABC analysis, now described, is not using proper population genetic simulations of allele frequency change per generation due to drift + selection. As written there is only one generation of drift. This needs to be changed to take this analysis even moderately seriously. Moreover the code for these simulations needs to be shared.

We have removed this analysis from the manuscript.

11) Line 1020-In the corcordance regression the authors are doing a binned regression-this is never appropriate and the authors need to redo this analysis without binning.

We agree that the binned regression would not be appropriate if we were solely interested in the statistical significance of the slope. However, we use the slope as a summary statistic to quantify the rate of change of concordance across the bulk of the genome.

Reviewer #3:Overall this is an improved manuscript. Easier to read and follow, and better explained. There are several points I think that should still be addressed.I am still not a fan of the flipped model. I agree that some of the evidence (predicting into the validation set, etc.) does indeed argue it's a better fit, but it still feels like ad-hoc subjective tweaking of the data until it fits well. I would prefer it to be removed from the paper -- I think show the original model and point out that some population show the reverse pattern and that matches with temperature. Perhaps even include the flipped model in the supplement. I would find that more convincing than the flipped model I think. In either case, the paragraph starting on line 445 should go, as even the authors admit this doesn't really show anything meaningful. The flipped model should also be removed from figures 2A and 2B as again it will by definition show a more convincing signal here.

We agree and have removed the flipped model. We hope you will find the analysis crisper and easier to follow.

As an alternative to the flipped model or presenting the data with the course labels of spring and fall, why not actually model the temperature data available? It would seem an objective a priori approach that should allow for differences in the flipped populations (i.e. presumably the difference in temperature the 3 weeks prior between Fall and Spring behaves differently for the flipped populations). Perhaps use mean temperature in the 3 weeks prior and/or the slope of the change of temperature over that time? I'm sure there are more creative/intelligent options, but I don't quite understand why the authors can't use this data instead of grossly categorizing things as spring or fall. I didn't see a good reason for not doing so in the response?

We approached the analysis of this dataset with a clear hypothesis that flies collected in spring (or fall), across a portion of the species range spanning and spanning two continents, would show allele frequency shifts at a common set of loci. While we find broad support for this hypothesis, we also find that it is not the whole story and provide evidence that aspects of weather in the weeks prior to sampling are associated with the direction of allele frequency change at a subset of loci. We agree with the reviewer that there are a number of different ways of characterizing environmental change across space and time. Reviewer 1 brought up a similar point. Indeed, there are an infinite number of ways of characterizing the environment. We explored a simple characterization (spring and fall), and extended this analysis to include some aspects of temperature as i) we have data about it, ii) it tends to be correlated with many other aspects of the environment, and iii) we have prior knowledge that temperature fluctuations of the kinds we consider are physiologically and developmentally relevant to *D. melanogaster*. We look forward to investigating the question of the environmental drivers of seasonal adaptation in future work that uses much more dense temporal and spatial sampling of flies. Importantly, we investigated a range of models and showed that only one set of models can explain the data. In addition, none of the nuisance variables – such as contamination with the D. simulans reads, or collection method, or geographic location – are correlated with the concordance statistic.

I find the authors treatment of enrichment odd. In some places it is presented as convincing evidence, and in others (line 563) it is disregarded because of absolute numbers. The logic on line 563 is fine of course, but I would like to see enrichment treated the same way throughout. On line 477 it is convincing as a log odds score, and in the paragraph starting on line 318 a modest percentage enrichment is considered good evidence.

We have streamlined the enrichment analyses and now present enrichment results solely for overlap with previously identified seasonal SNPs: “Top 1% of SNPs identified using the GLM overlap with SNPs identified by Bergland et al. (2014) slightly more than expected relative to matched genomic controls after re-identifying seasonal SNPs in the Core20 set excluding the overlapping Pennsylvanian populations (log_2_ odds ratio ± SD = 0.59 ± 0.37, p_perm_ = 0.0512).”

I'd like to see a bit more exploration of the clustering. Figure 5D (visually) and the 100kb window analysis seem to suggest that clustering is on a relatively large scale, yet the analysis presented on 629 for % genome and s only investigates 5kb windows. If I'm understanding the ABC correctly it should be pretty fast to run, and it seems like running it on 50, 100, or even 500kb scale might be of interest. (To my eye some of the figures in S10 start to suggest a flattening of the ridge when done at 5kb scale). Certainly the data do appear to argue for a polygenic architecture, but whether this is ~50 windows or 5% of the genome I think isn't well differentiated.

We have removed this analysis from the manuscript.

Line 800: I agree with this logic about temperature and why some populations behave differently. I would have liked to see this prediction about temperature earlier in the introduction. Naively my first impression was that Fall populations would be adapted to cooler conditions and Spring to warmer. I see now why that is wrong, but I think stating up front that Fall populations are expected to reflect adaptation to warm summers would help some readers.

We now explain in the introduction that flies collected in spring are the descendants of flies who survived winter, and that flies collected in fall are the descendants of those lineages that prospered during the summer.

[Editors’ note: further revisions were suggested prior to acceptance, as described below.]

Almost there! As you will see from the comments below, we got a fresh 3rd reviewer, who picked up on something we agree should be addressed, namely the likely role of inversions. We have two suggestions for how to proceed. Either: a) go through the manuscript and make sure you emphasize that much of the signal may be driven by inversions, and that it is impossible to know how polygenic this really is, or; b) provide additional analyses (e.g., of the X chromosome), to demonstrate that there is a signal independent of inversions.

We have edited the text to emphasize that the signal may be driven largely by inversions and that the extent of polygenicity is difficult to ascertain with the current dataset.

Reviewer #3:The strongest seasonal signal comes from inversions. If inversions are responding to seasonal selection, it is not surprising that the authors find parallel SNP changes across populations as the same inversions are shared globally. Unless the authors refocus the manuscript on parallel selection on inversions, their analyses need to be modified: almost all analyses use the full SNP set, but to study real parallel selection responses on the SNP level, the authors need to restrict their analysis on SNPs, which are not affected by inversions. To this end, it is important to keep in mind that inversions may suppress recombination also outside of the inversion, which makes it a bit challenging to determine the autosomal fraction that is not affected by inversions. A much better strategy would be to analyze the X-chromosome, which is the only major chromosome free of inversions. Unfortunately, the authors excluded this chromosome from their analyses.

We appreciate the reviewer’s suggestion about analyzing the X-chromosome. We excluded the X-chromosome in our analysis because we sequenced males, and thus have reduced power to detect seasonal allele frequency shifts. The reduction of power on the X compared to the autosomes would complicate any direct comparison. Therefore, we have taken your suggestion (and that of the editors) and refocused the manuscript on parallel selection on inversions.

Line 108: removed the word “polygenic”.

Lines 207-210: added statistics on the seasonal enrichment outside inversions.

Line 349: Removed the claim: “While we cannot determine the precise number and identity of causal sites in the present study, the totality of the evidence is consistent with the polygenic nature of adaptation.”

Lines 352-355: comment on the power to detect parallel seasonal evolution in low recombination regions, such as inversions.

Lines 359: changed “strongly polygenic” to “complicated architecture”.

Line 412: added mention of selection on inversions obscuring the estimate of number of seasonal loci.

Lines 420-422: added “While given our dataset it is impossible to know the true extent of polygenicity, our results suggest that seasonal adaptation acts on at least several but potentially a large number of dispersed loci affecting linked variation genome-wide.”

Line 426. changed the claim of seasonal adaptation being polygenic to a claim of seasonal adaptation at common SNPs.

Title: changed to “Broad geographic sampling reveals the shared basis and environmental correlates of seasonal adaptation in *Drosophila*”

Anyway, inspection of Figure 2D shows that the signal for seasonal SNPs is erased for regions outside of the inversions. Furthermore, a significant concordance pattern between seasonal and clinal SNPs outside of the inversion is restricted to 2L and 3R, the chromosomes with the strongest inversion effects. This could be interpreted as an effect of inversions on the genomic regions flanking the inversion.

The reviewer is correct that seasonal SNPs do not appear to be enriched outside of regions surrounding inversion breakpoints or inside of inversions on each of the autosomal arms (see added statistics, lines 207-210). As we pointed out above, we have refocused our manuscript to highlight the likely role of inversions associated with seasonal adaptation in *D. melanogaster*. However, we caution that inversions are not solely responsible for seasonal adaptation and speculate that the true mechanistic dynamics are likely more complicated.

How do the authors interpret a (presumably significant) underrepresentation of concordance SNPs on 3L?

The under-enrichment in concordance of seasonal and clinal polymorphisms outside of the inversion on 3L is not statistically significant after multiple testing correction (FDR=0.69). However, for the sake of speculation, it is possible that counter gradient evolution can occur at certain SNPs, wherein alleles that are beneficial over the winter are also higher frequency in the south compared to the north. Future studies with larger sample sizes will be necessary to assess whether there truly is a consistent inverted pattern in this genomic region.

Apart from my doubts about the significance of the seasonal selection signal, I would like to come back to the novel aspect of the manuscript-sharing seasonal SNPs across populations. The authors highlight, probably correctly, that seasonal adaptation is polygenic. This raises the question of whether parallel selection signatures are expected in differentiated populations. In my opinion two lines of reasoning speak against it: 1) probably more variants are segregating in the populations than required for seasonal adaptation (redundancy) 2) the frequencies of the seasonal SNPs most likely differ between the populations. Hence, SNPs closer to 50% are expected to respond more to the same selection pressure than SNPs with more extreme allele frequencies. This will lead to different power to detect the same selection response in differentiated populations.

These are completely reasonable points. In the discussion, we now highlight how redundancy could affect the seasonal signal that we observe. We believe that point #2 is reasonable but addressing this question rigorously is outside of the scope of the present manuscript.

Analyze the X-chromosome.

Please see the comments above.

Remove the second season from the locations where two spring-fall pairs were included-only this makes the comparison unbiased.

While we appreciate the concern here, we think that it is unfounded. As shown in Bergland et al. 2014, population differentiation between years is as strong – on average – as populations separated by 5° latitude. Although each locality in our sample seems to represent a stable population (i.e., no mass extirpation), it is not clear that having two years of samples from a single locality is any more biased than having seasonal samples from neighboring localities (e.g., Linvilla PA and State College PA). In addition, we show that there is a significant excess of seasonal sites using the BayEnv model which accounts for genetic covariance among samples.

Evaluate whether the spring-fall permutations remove the statistical issues of the GLM and Fisher's exact tests mentioned by the previous reviewers. Clarify that the matched controls were done on a sample basis, rather than across samples.

The 100 matched control datasets were generated with the characteristics of each SNP – not by using any single-population characteristics. One potential reason this could have been misinterpreted is that one of the matching characteristics is median spring allele frequency; this being the median across populations. We have clarified this in the methods (lines 585-586).

Clarify that the effective coverage was calculated per SNP.

We have clarified this in the methods (line 531).

The authors cite theoretical work, which suggests that seasonal SNPs may be maintained for highly restricted conditions (changing dominance)-do they find empirical support that these conditions are met in their data?

We are unable to test for changes in dominance coefficients between seasons using the data that we present in this manuscript.

The significance of the manuscript to a broader audience could be increased by:– A statement that the seasonal selection response is restricted to inversions-but I doubt that this is the message the authors would like to portray.

Please see the comments above.

– A general discussion about the expectations of parallel selection signatures on the SNP level across populations and why the authors expect to see it (or find it against the expectations).

Please see the comments above.